

# Upper tropospheric pollutants observed by MIPAS: geographic and seasonal variations

Norbert Glatthor[1], Gabriele P. Stiller[1], Thomas von Clarmann[1,†], Bernd Funke[2], Sylvia Kellmann[1], and Andrea Linden[1]

[1]Karlsruhe Institute of Technology, Institute of Meteorology and Climate Research, Karlsruhe, Germany
[2]Instituto de Astrofísica de Andalucía, CSIC, Granada, Spain
[†]deceased, 13 January 2024

**Correspondence:** N. Glatthor (norbert.glatthor@kit.edu)

**Abstract.** We present a global climatology of upper tropospheric hydrogen cyanide (HCN), carbon monoxide (CO), acetylene ($C_2H_2$), ethane ($C_2H_6$), peroxyacetyl nitrate (PAN) and formic acid (HCOOH), obtained from MIPAS/Envisat observations between 2002 and 2012. At northern mid- and high latitudes the biomass burning tracer HCN as well as CO, PAN and HCOOH exhibit maxima during spring and/or summer and minima during winter. On the contrary, maximum northern extra-tropical

$C_2H_2$ and $C_2H_6$ amounts were measured during winter and spring and minimum values during summer and fall. In the tropics and subtropics, enhanced amounts of all pollutants were observed during all seasons, especially widespread and up to southern mid-latitudes during austral spring. Other characteristic features are eastward transport of anthropogenic $C_2H_6$ and of biogenic HCOOH from Central and North America in boreal summer, accumulation of pollutants in the Asian Monsoon Anticyclone and enhanced $C_2H_2$ over South-East Asia in boreal winter. Clear indication of biogenic release of HCOOH was also found above

tropical South America and Africa. A global correlation analysis of the other pollutants with HCN corroborates common release by biomass burning as source of the widespread southern hemispheric pollution during austral spring. Further, high correlation with HCN points to biomass burning as major source of tropical and subtropical $C_2H_2$ and PAN during most of the year. In the northern extra-tropics there are generally low correlations with HCN during spring and early summer, indicating the influence of anthropogenic and biogenic sources. However, in August there are stronger correlations above Siberia and

boreal North America, which points to common release by boreal fires. This is confirmed by the respective enhancement ratios (ERs). The ERs measured above North-East Africa fit well to the emission ratios of the dominant local fire type (savanna burning) for $C_2H_2$, while those for CO, $C_2H_6$ and HCOOH rather indicate tropical forest fires or additional anthropogenic or biogenic sources. The southern hemispheric $\Delta C_2H_6/\Delta HCN$ ERs obtained during August to October are in good agreement with the emission ratio for savanna fires. The same applies for $\Delta C_2H_2/\Delta HCN$ in August and for $\Delta HCN/\Delta CO$ as well as for

$\Delta HCOOH/\Delta HCN$ in October.

## 1  Introduction

Hydrogen cyanide (HCN) is considered as an almost unambiguous tracer of biomass burning (Li et al., 2003; Singh et al., 2003; Yokelson et al., 2007; Lupu et al., 2009), while the other trace gases presented here generally have additional sources,



too. HCN has a tropospheric lifetime of 5–6 months, which is controlled by ocean uptake (Li et al., 2000, 2003; Singh et al., 2003). Carbon monoxide (CO) is released from traffic, industry and biomass burning by incomplete combustion processes and is a precursor of tropospheric ozone. However, approximately one half of its tropospheric burden originates from in situ photochemical production by oxidation of methane and more complex hydrocarbons (Park et al., 2013). Because of its

relatively long lifetime of about 2 months, CO has often been used as tracer for transport of tropospheric pollution (Rinsland et al., 1998; Edwards et al., 2006; Funke et al., 2009). Its upper tropospheric concentrations are generally much higher than those of the other trace gases presented here. Acetylene ($C_2H_2$) is a relatively reliable but not unique tracer of biomass burning (Blake et al., 1996; Singh et al., 1996). More recent studies (Streets et al., 2003; Xiao at al., 2007) indicate biofuel emissions from cooking fires or residential heating as its dominant source. Xiao et al. (2007) assume a global biofuel source of 50% and

global fossil fuel and biomass burning sources of 25%, respectively. The upper tropospheric lifetime of $C_2H_2$ is about 2 weeks at low latitudes or during extra-tropical summer (Xiao et al., 2007). During winter, its lifetime increases considerably at higher latitudes due to the absence of the hydroxyl radical (OH) under less sunlit conditions. Ethane ($C_2H_6$) is the most abundant tropospheric non-methane hydrocarbon (Singh et al., 2001). Rudolph (1995) indicates biomass burning and natural gas losses as its major sources, with releases of 6.4 and 6 Tg yr$^{-1}$, respectively. According to Xiao et al. (2008) the major source of $C_2H_6$

is fossil fuel production (62%), followed in about equal parts by biofuel and biomass burning. Its mean atmospheric lifetime is approximately 2 months (Hough, 1991), but 6 to 10 months in the extratropical upper troposphere (UT) during winter (Xiao et al., 2008; Helmig et al., 2016). Peroxyacetyl nitrate (PAN, $CH_3C(O)OONO_2$) is formed via oxidation of hydrocarbons and subsequent reaction with $NO_2$, and thus generated in air masses polluted by fuel emissions or by biomass burning (Singh et al., 1996; Gaffney et al., 1999). Fischer et al. (2014) emphasize the importance of lightning $NO_x$ for the production of

PAN. Further, according to these authors the most important precursor of PAN is isoprene (37%), which is mainly released from plant foliage (Guenther et al., 2006). Other precursors are acetone (9%) and ethane (6%). The lifetime of PAN is highly temperature dependent and can be up to several months in the cold upper troposphere (Singh, 1987). It is the major reservoir of tropospheric $NO_x$ (NO and $NO_2$) and enables the transport of $NO_x$ to remote tropospheric areas. Direct sources of formic acid (HCOOH) are biogenic release, anthropogenic emissions (biofuel or fossil fuel burning) or biomass burning (Grutter et al.,

2010, and references therein). While Grutter et al. (2010) state that the relative contribution of direct emissions and chemical transformations is unknown, oxidation of hydrocarbons is declared as the dominant source of atmospheric HCOOH in other publicatons (Sanhueza et al., 1996, and references therein). According to Stavrakou et al. (2012), the contribution of biogenic sources (tropical and boreal forests) to atmospheric HCOOH is as high as 90%. Franco et al. (2020) state strong biogenic sources in tropical spring and northern extra-tropical spring and summer.

In the following, we consider observation of enhanced HCN amounts as an almost unique indication of pollution by biomass burning. A high correlation of any other trace gas with enhanced HCN points to biomass burning as important source of this pollutant as well. Measurement of high $C_2H_2$ volume mixing ratios (VMRs) in combination with low HCN amounts hints at biofuel or fossil fuel burning. Exclusive measurement of high $C_2H_6$ amounts hints at release during production or transport of oil and gas. Measurement of enhanced PAN or HCOOH not correlated with elevated HCN points to biogenic sources or

anthropogenic emissions other than biomass burning of these trace gases.





Two other spaceborne data sets containing most of the pollutants presented here result from measurements of ACE-FTS on Scisat (Bernath et al., 2005; Boone et al., 2005) and of IASI on Metop-A (Blumstein et al., 2004; Hilton et al., 2012). ACE-FTS is a limb sounder performing solar occultation measurements in the spectral range of 750–4400 $cm^{-1}$. ACE-FTS measurements of HCN have been presented by Lupu et al. (2009). Glatthor et al. (2015) performed a comparison between HCN observed by

MIPAS and by ACE-FTS. González Abad et al. (2009) discuss seasonal latitude-height cross sections of HCOOH, González Abad et al. (2011) of CO, $C_2H_6$ and $C_2H_2$, and Tereszchuk et al. (2013a) of PAN observed by ACE-FTS. A climatological analysis of the trace gases CO, $C_2H_6$, $C_2H_2$ and HCN in the upper troposphere and lower stratosphere has been performed by Park et al. (2013), but focused on the tropics only. IASI is a nadir sounder covering the infrared spectral region from 645 to 2760 $cm^{-1}$, which carries out cross-track scans with a swath-width of ∼2400 km. Observations of tropospheric trace gases

by IASI are generally presented as column amounts. Spatial distributions of CO observed by IASI have been presented by Clerbaux et al. (2009), tropical and subtropical HCN and $C_2H_2$ of IASI has been shown by Duflot et al. (2015), and HCOOH by Pommier et al. (2016) and Franco et al. (2020).

In the following we will shortly describe the MIPAS instrument, the retrieval setups and the method of tropopause calculation. Then we will present seasonal global distributions of the pollutants on a surface at 3 km below the tropopause. Further,

we will discuss the monthly variation of zonally averaged time series. By means of global correlation analyses with HCN we will discriminate between biomass burning and other anthropogenic or biogenic sources of the individual pollutants. The enhancement ratios (ERs) resulting from the correlation analyses will be compared with enhancement ratios obtained from ACE-FTS measurements and with emission ratios derived from published emission factors. Our presentation will be restricted to seasonal and monthly composites of the whole MIPAS data set. Interannual variations of, e.g., MIPAS HCN have been

described by Glatthor et al. (2015). Exemplarily, some monthly distributions of correlation coefficients and ERs obtained for individual years will be presented in the Supplement. The paper will be finished with a summary and conclusions.

## 2 MIPAS measurements

### 2.1 Instrument description

The Michelson Interferometer for Passive Atmospheric Sounding (MIPAS) was operated onboard the European polar-orbiting

ENVIronmental SATellite (ENVISAT) from July 2002 until April 2012. MIPAS was a Fourier transform infrared (FTIR) emission spectrometer with a measurement range from 685 to 2410 $cm^{-1}$, which enabled the observation of numerous trace gases emitting in the mid-infrared spectral region (European Space Agency (ESA), 2000; Fischer et al., 2008). Due to its measuring principle MIPAS was able to observe the atmosphere in global coverage during day and night.

Until March 2004 MIPAS was operated with the spectral resolution of 0.025 $cm^{-1}$ (unapodized) in the so-called full resolu-

tion (FR) mode. After a data gap due to technical problems during the rest of the year 2004, MIPAS was run with the spectral resolution of 0.0625 $cm^{-1}$ in reduced resolution (RR) mode since January 2005. We present data of the FR and RR nominal modes and of the RR Upper Troposphere and Lower Stratosphere 1 mode (UTLS-1) with horizontal sampling distances of 510, 410 and 290 km, respectively. The nominal modes consisted of rearward limb-scans covering the altitude ranges 6–68 km (FR)





and 7–72 km (RR). The UTLS-1 mode was restricted to the altitude region 5.5–49 km. The tangent height distance of the FR mode was 3 km up to 42 km and increased up to 8 km between the two uppermost tangents. The step-width of the RR nominal mode was 1.5 km up to 22 km and then increased to 2, 3 and 4-5 km in the upper part of the scan. The step-width of the RR UTLS-1 mode was 1.5 km up to 19 km, 2 km up to 25 km and increasingly wider above.

## 2.2 Retrieval method and error estimation

Retrievals were performed with the processor of the Institute of Meteorology and Climate Research (IMK) and the Instituto de Astrofísica de Andalucía (IAA) by use of level-1B radiance spectra of data version 5.02/5.06 (reprocessed data) provided by the European Space Agency (ESA) (Nett et al., 2002). The central components of the retrieval processor are the Karlsruhe Optimized and Precise Radiative Algorithm (KOPRA) (Stiller, 2000) for radiative transfer calculations and the Retrieval Control Program (RCP) of IMK/IAA for inverse modelling. The inversion consists in determination of vertical profiles of atmospheric state parameters by constrained non-linear least squares fitting in a global-fit approach (von Clarmann et al., 2003). The general approach of MIPAS data processing at IMK has been described in various papers, e.g., in von Clarmann et al. (2003), Höpfner et al. (2004) or von Clarmann et al. (2009).

Retrieval of the trace species presented here has been described in previous papers (Glatthor et al., 2007, 2009, 2015, von Clarmann et al., 2009, Funke et al., 2009, Grutter et al., 2010, Wiegele et al., 2012). The data versions used in this presentation are compiled in Table 1. Since the altitude spacing of the retrieval grid chosen is finer than the height distance between the tangent altitudes, potential instabilities have to be attenuated by application of a constraint. For this purpose, Tikhonov's first derivative operator was used (Tikhonov, 1963; Steck, 2002). Instead of climatological a-priori profiles, height-constant profiles were chosen to avoid any influence of the a-priori information on the shape of the retrieved profiles. Generally, profiles of the main interfering trace gases were jointly fitted along with the target species. Additional retrieval variables were microwindow-dependent continuum radiation profiles and microwindow-dependent, but height-independent zero-level calibration corrections. If available, the radiative contribution of other interfering gases was modelled by using their profiles retrieved in preceding steps in the processing sequence. When no prefitted profiles were available, we used the climatology compiled by Remedios et al. (2007) for this purpose. MIPAS measurements provide information on the trace gases presented here from the free troposphere up to about 25 km for $C_2H_2$, $C_2H_6$ and PAN, up to about 45 km for HCN, and up to 70 km (nominal mode scans) for CO. The vertical resolution of the different pollutants in the altitude region displayed in the following sections (2–3 km below the tropopause, which dependent on latitude corresponds to geometrical altitudes of ∼8–14 km) is displayed in Table 2. At 8 km in the northern extra-tropics it varies between 2.3 km (PAN) and 4.3 km (HCN) and at 14 km in the southern hemispheric tropics and subtropics between 2.8 km (PAN) and 4-4.5 km (HCN, CO, $C_2H_6$). Table 2 also contains the estimated standard deviation (ESD), which is later used to determine the $\lambda$-values in the regression analyses. The total retrieval errors for significantly enhanced VMRs of these pollutants in the upper troposphere are between 5 and 15%. More details on error estimation and vertical resolution are given in the publications mentioned above.





# 3 Tropopause height derived from MIPAS temperature profiles

To enable a tropopause-related presentation of the pollutants, we calculated the thermal tropopause for each MIPAS scan from the temperature profile retrieved in an earlier step of the processing chain. The calculation of the thermal tropopause was performed according to the definition of the World Meteorological Organisation (WMO). In this definition the tropopause is

the lowest level, at which the lapse rate $-dT/dz$ decreases to 2 K km$^{-1}$ or less and does not exceed this value in an overlying layer of the width of at least 2 km. Technically, we determined the tropopause as described in Zängl and Hoinka (2001, their Eq. A1). This means, lapse rate profiles were calculated as centered difference quotients between each two adjacent retrieval heights. Then the lowest lapse rate level with $-dT/dz \leq 2$ K km$^{-1}$ was searched. The tropopause height was determined by linear interpolation between the lapse rate at this level and the level below to the 2 K km$^{-1}$ value, provided the criterion for

layer width was fulfilled. If the latter criterion was not met, the next level with $-dT/dz \leq 2$ K km$^{-1}$ was checked. In a subsequent step the retrieved profiles of the pollutants were related to the individual tropopause heights.

Figure 1 shows geographical distributions of seasonal tropopause heights averaged over the whole MIPAS dataset. The tropical tropopause height is about 16.5 km during summer and fall and around 17 km in winter and spring. Around 30-40°N and 30-40°S the tropopause height decreases strongly, reaching a level of 8–9 km at high northern and of 8–12.5 km at high

southern latitudes. Compared to the equinoxes, the region of the strongest latitudinal gradient is shifted northward during boreal summer and southward during austral summer. Beside the distinct latitude dependence, the tropopause heights obtained from MIPAS data exhibit a longitudinal variation as well. The most obvious feature is a variation during boreal winter with low tropopause heights above Eastern Canada and East Siberia and a high tropopause above the North Atlantic and Northern Europe, which is connected with oscillations of the polar jet.

At latitudes south of 60°S the average tropopause height is considerably higher during austral winter (JJA) and spring (SON) than in the remaining part of the year or in the Arctics during boreal winter and spring. This is caused by temperature profiles, for which the WMO criterion was not met in the lower atmosphere due to unusually high lapse rates, leading to determination of tropopause altitudes of up to 20 km. For illustration we show the individual tropopause heights obtained for the different seasons of the year 2010 in the Supplement (Fig. S1). The disappearance of the antarctic thermal tropopause during the final

months of austral winter (July/August) has first been reported by Court (1942). Similar problems have been reported by, e.g., Reichler et al. (2003). Following the approach of Zängl and Hoinka (2001), tropopause heights of more than 14.7 km ($\sim$90 hPa) were fixed to this value in the latitude band south of 60°S to reduce the effect of this artefact. Since quite a number of unplausible high tropopause heights were also obtained for antarctic spring and arctic winter (Fig. S1), the same criterion was generally applied to tropopause heights at higher latitudes than 60° during all seasons.



## 4 Results and discussion

### 4.1 Global distributions at 3 km below the tropopause

In the following we present global distributions of seasonal composites of the different pollutants at 3 km below the tropopause. This distance was chosen to be sure that, related to the vertical resolution of MIPAS measurements, the distributions are nearly

completely tropospheric. The data shown are bin averages (5° lat × 15° lon, 7.5° lat × 15° lon at the poles) of the whole measurement period of MIPAS (2002–2012). The individual bins typically contain 100 to more than 600 data points at low- and mid-latitudes, but generally much lower amounts (dozens to less then 10) at high latitudes. During strong convection the tropical bins over the continents contain considerably lower amounts of data points, too.

### 4.1.1 HCN (Hydrogen cyanide)

During March to May, the climatological HCN distribution exhibits widespread enhancements with mixing ratios of up to 350 pptv at northern mid- and partly high latitudes (Fig. 2a). Extensive biomass burning at northern mid-latitudes during boreal spring is indicated by fire carbon emissions of the Global Fire Emissions Database, version 3.1 (GFEDv3.1, Randerson et al., 2013), added up over the measurement period of MIPAS (Figure 3, top left). Strong carbon emissions extend over a large area from Eastern Europe over Southern to East Siberia. According to van der Werf et al. (2010, their Fig. 13), the dominant fire types

in this region are agricultural fires in Eastern Europe and forest fires in South-East Siberia. A second region with enhanced carbon emissions in boreal spring extends from the central United States to southern Canada. Other northern hemispheric source regions are South- and South-East Asia, Central America and tropical Northern Africa. Using trajectory calculations of the Hybrid Single-Particle Lagrangeian Integrated Trajectory Model (HYSPLIT) of the National Oceanic and Atmospheric Administration (NOAA) (Stein et al., 2015) and fire counts of the Moderate Resolution Imaging Spectroradiometer (MODIS)

(Justice et al., 2002) on the Aqua and Terra satellites, individual MIPAS HCN plumes observed at northern mid-latitudes during boreal spring could clearly be traced back to sources in North-East Africa, South-East Asia and Central America (Figs. S2-S4 in the Supplement). Although such clear examples for release of discrete HCN plumes from e.g. eastern Europe or southern Siberia could not be found in the samples analysed so far, a significant contribution of these regions to pollution of the northern extra-tropical UT by HCN during boreal spring can not be excluded.

The highest northern hemispheric HCN amounts, now also covering the whole Arctic, were measured during June to August (Fig. 2c), with maxima of up to 430 pptv above Siberia and the Canadian Arctic. During this season, the strongest NH GFED carbon-emissions are located in Siberia and in Canada/Alaska (Fig. 3, top right), which can be associated to boreal forest fires (van der Werf et al., 2010, their Fig. 13). Other potential source regions are Eastern Europe, the Mediterranean area and the North-Western United States, but contributions from the northern tropics and subtropics have strongly decreased. As examples

for the transport of HCN from a fire region in Western Russia into the polar upper troposphere and for the distribution of enhanced HCN from another region in North-East Siberia over the northern hemisphere we present MIPAS HCN measurements from August 2008 and 2010 as well as respective HYSPLIT trajectories and MODIS fire counts in the Supplement (Figs. S5, S6).



After the end of the fire season the HCN amounts in the northern hemisphere continuously decline during boreal fall and winter, with especially low amounts over the northern Atlantic and Europe. According to the current state of knowledge (Li et al., 2000, 2003; Singh et al., 2003) low HCN amounts over oceanic regions show the removal of HCN by ocean uptake. During boreal winter, the HCN VMRs above the Arctic Ocean also have decreased to background levels, while moderately enhanced

HCN amounts are located above North-Eastern China, the North Pacific Ocean and Northern Canada. These enhancements can be remains of the preceding boreal fire season, but ca also be caused by residential coal burning in East-Asia as suggested by Singh et al. (2003) and Li et al. (2003).

In the tropics, enhanced HCN amounts were observed above or around Africa during all seasons, which is in line with year-round strong fire activity on this continent. The plume above Central and Northeast-Africa during March to May, extending

eastward towards India is caused by biomass burning in the savanna region south of the Sahara, which is confirmed by GFED carbon emissions (Figure 3, top left). Actually, this process already peaks in boreal winter (see Fig. 3, bottom right), but pollutants in this area are more effectively uplifted by increased convection during boreal spring (Duncan et al., 2007; Glatthor et al., 2015). During all seasons, the lowest tropical HCN values (175 – 200 pptv) were measured above the Pacific due to ocean uptake.

In boreal summer, a HCN plume with VMRs of up to 350 pptv extends from the Arabian peninsula to East China. This feature is caused by confinement of pollution mostly from South- and South-East Asia (see, e.g., Vogel et al., 2015) inside the Asian Monsoon Anticyclone (AMA). The slightly lower enhancements over North Africa and the Northern Pacific reflect the climatological average of east- and westward outflow.

During September to November, biomass burning has shifted towards the southern hemisphere (SH), characterized by

strongly enhanced HCN amounts in a large region extending from tropical Brazil over southern Africa to the region south of Australia and to the South-Pacific. The latitude band between 30 and 60°S is nearly completely covered with enhanced HCN. Correspondingly, the GFED data show strong fire activities in southern tropical and subtropical South America and Africa (Fig. 3, bottom left). There is also indication of another southern hemispheric plume above the eastern tropical Indian Ocean at 15°S. This feature is caused by strong biomass burning in Indonesia during the years 2002,

2006 and 2009 (cf. Glatthor et al., 2015) with a positive phase of the so-called El Niño-Southern Oscillation (ENSO) (https://www.cpc.ncep.noaa.gov/products/precip/CWlink/MJO/enso.shtml), but attenuated by averaging over all years of MI-PAS operation. Strong biomass burning in Indonesia as well as in northern Australia during this season is confirmed by the GFED carbon emissions (Figure 3, bottom left). In Fig. S8 of the Supplement we present HYSPLIT forward trajectories started above the biomass burning regions in South America, southern Africa and Indonesia, which corroborate the distribution

of polluted air masses in the southern hemisphere as observed in the MIPAS data.

During December to February the pollution in the southern tropics and subtropics has considerably decreased, and enhanced HCN amounts are mostly restricted to the southern Atlantic and southern Africa. Howver, due to the time-lagged southward transport of the polluted air masses, maximum HCN amounts of about 270 pptv were measured at high southern latitudes. Similar observations were made by Zeng et al. (2012). In the following two seasons (March to August) the HCN amounts at

southern mid- and high latitudes continuously decrease to 150 pptv, which we also attribute to ocean uptake.



### 4.1.2 CO (Carbon monoxide)

At northern mid- and high latitudes, the highest upper tropospheric CO amounts of up to 120 ppbv were observed during boreal spring and summer (Fig. 2b, d). This similarity to the HCN cycle points to biomass burning as important source of enhanced CO. However, the CO amounts in the NH are elevated during the whole year, resulting in a weaker annual variation. This indicates additional anthropogenic sources as well.

Enhanced CO amounts were also found in the tropics and subtropics all the year round in the same regions like elevated HCN, indicating common release by biomass burning. The most prominent signatures in these regions are a plume covering large parts of Africa and extending to South-Asia and the northern subtropical Pacific during March to May, a plume inside the Asian Monsoon Anticyclone during June to August and the southern hemispheric biomass burning plume during September to November. Different to HCN, enhanced CO covers South- and Southeast Asia and the Eastern Pacific during the whole year. This hints at a significant fraction of CO from other anthropogenic sources, e.g. in South- and East-Asia, as well.

Much lower CO VMRs were observed in the southern extra-tropics. Caused by southward transport of air masses polluted by biomass burning at southern low latitudes, the CO amounts at southern mid- and high latitudes exhibit an annual cycle with moderately enhanced values in austral spring, but low values in austral summer and fall. Thus, different to the longer-lived HCN, the CO amounts at high southern latitudes do not further increase in austral summer, but decline instead. Again, this is consistent with observations of Zeng et al. (2012).

### 4.1.3 $C_2H_2$ (Acetylene)

The $C_2H_2$ amounts at northern mid- and high latitudes are considerably higher than those observed in the tropics and at southern latitudes (Fig. 4, left column). Different to HCN, the northern hemispheric $C_2H_2$ cycle has a maximum of up to 250 pptv during winter and spring and a minimum of 40–80 pptv during summer. This indicates that biomass burning is not the reason of these variations. According to Zander et al. (1991), the accumulation of $C_2H_2$ in winter and spring is due to the wintertime minimum of the hydroxyl radical OH, which is the major reactant for depletion of tropospheric $C_2H_2$ (Rudolph et al., 1984). An additional cause is increased domestic heating of biofuel in winter (Streets et al., 2003). During all seasons, but especially during winter and spring, the northern extra-tropical $C_2H_2$ amounts show a longitudinal variation, characterized by lower values above the North Atlantic and Western Europe and higher values above the Canadian Arctic and Eastern Siberia. This feature is correlated with the longitudinal oscillations in tropopause height visible in Figure 1 and of the polar jet, leading to northeastward transport of air masses with low $C_2H_2$ amounts above the Northern Atlantic.

The tropical and subtropical $C_2H_2$ distribution with maximum amounts of up to 120 pptv over Africa and up to 140 pptv in the AMA is spatially well correlated with HCN during all seasons. This points to biomass burning as dominant source of the enhanced $C_2H_2$ amounts observed at these latitudes. One exception is South- and South-East-Asia during winter, where at lower altitudes around 11 km elevated VMRs of $C_2H_2$ were observed in combination with low HCN amounts (Fig. A1). This hints to increased biofuel burning during this season. Similar to HCN, the most widespread southern tropical $C_2H_2$ pollution, extending from North-East Brazil over Southern Africa to Australia, occurs during September to November.



The $C_2H_2$ mixing ratios at southern mid- and high latitudes vary between less than 20 pptv during austral summer and fall (December–May) and 40–50 pptv during austral winter (JJA). Shifted by half a year, this is a similar seasonality like in the NH, indicating OH control as well. During austral spring there are slightly enhanced $C_2H_2$ amounts of 50–80 pptv between 30°S and 50°S, which reflects southward transport of air masses polluted by southern hemispheric biomass burning. Due to its
relatively short lifetime, southward expansion of enhanced $C_2H_2$ to antarctic latitudes in austral summer is not observed.

### 4.1.4  $C_2H_6$ (Ethane)

Similar to $C_2H_2$, much higher $C_2H_6$ amounts were observed in the northern extra-tropics than in the tropics and at southern latitudes (Fig. 4, right column). The annual cycle at northern mid- and high latitudes with maximum VMRs of up to more than 1000 pptv in winter and spring and minimum values around 700 pptv in late summer and fall shows that these variations are
also not dominated by biomass burning. Like for $C_2H_2$, the annual cycle of the OH radical is the dominant reason for the $C_2H_6$ maximum in the NH during winter (Rudolph, 1995; Xiao et al., 2008; Helmig et al., 2015). For the same reasons as outlined above, the northern extra-tropical $C_2H_6$ distribution shows similar longitudinal variations as $C_2H_2$ during fall and winter.

The tropical and subtropical $C_2H_6$ distribution exhibits similarities to those of HCN and $C_2H_2$. There are also moderately enhanced $C_2H_6$ amounts of $\sim$500 pptv above West- and North-Africa in spring, inside the AMA in summer and in the SH in
austral spring, indicating release of $C_2H_6$ from biomass burning. However, during boreal summer, $C_2H_6$ amounts of 500-600 pptv are visible around Central America, above the South-Eastern United States and the North Atlantic. These enhancements, which are not correlated with HCN and $C_2H_2$, become even better visible further below the tropopause. Therefore they will be presented at 11 km altitude in the Appendix in comparison with the other trace gases.

The lowest $C_2H_6$ VMRs of 200 pptv or less were observed at southern mid- and high latitudes during austral summer (DJF)
and fall (MAM). In austral winter (JJA) the amounts increase slightly at southern mid-latitudes, which obviously is also due to the OH cycle. As a result of biomass burning in the southern tropics and subtropics during austral spring (SON) the $C_2H_6$ amounts accumulate somewhat more at southern mid-latitudes and even slightly increase at high latitudes. Contrary to HCN there is no further $C_2H_6$ enhancement but rather a reduction at high southern latitudes in austral summer (DJF), which probably is due to its shorter lifetime.

### 4.1.5  PAN (peroxyacetyl nitrate)

By far the highest PAN amounts were observed at northern mid- and high latitudes during boreal summer (Fig. 5c), with maximum amounts of 450 pptv above north-eastern Europe and Siberia. The lowest northern hemispheric PAN VMRs of 70–130 pptv were measured during boreal winter. The spatial distribution and seasonal variation of northern extra-tropical PAN exhibit similarities to HCN, indicating biomass burning as a potential source. However, according to Fischer et al. (2014)
PAN production in the northern extra-tropics is driven by anthropogenic and biogenic activities during spring and dominated by biogenic species during summer, with isoprene as major precursor (37%). Other non-methane volatile organic compounds listed by Fischer et al. (2014), which contribute to PAN formation, are acetone and $C_2H_6$. We will further investigate the origin



of northern hemispheric PAN in Section 4.3. Similar to $C_2H_6$, enhanced PAN amounts were also measured above the northern Atlantic between the East Coast of the United States and Europe.

The tropical and subtropical PAN distribution is spatially well correlated with HCN during all seasons, pointing to biomass burning as an important source of upper tropospheric PAN in these regions. During the southern hemispheric fire season in

austral spring (SON) PAN VMRs of up to 240 pptv were observed between South America and Southern Africa. In this season enhanced PAN VMRs of up to 150 pptv also cover the latitude band 30°S–50°S, reflecting southward expansion of air masses polluted by biomass burning. Similar to $C_2H_2$ and $C_2H_6$ there is no further expansion of enhanced PAN amounts to high southern latitudes in austral summer (DJF). During the other seasons the PAN amounts at southern hemispheric mid- and high latitudes are very low.

**4.1.6   HCOOH (Formic acid)**

The highest HCOOH VMRs were observed at northern mid- and high latitudes during summer, with maxima of 180 pptv above Siberia and the Canadian Arctic (Fig. 5d). During winter, the HCOOH amounts in the northern extra-tropics decline to 50–70 pptv (Fig. 5h). The correlation of the annual variation of northern hemispheric HCOOH both with the vegetation and HCN cycles points to biogenic sources as well as to release by biomass burning. This will be further investigated in Section 4.3.2.

The spatial distribution of enhanced HCOOH above tropical and subtropical Africa is in good agreement with enhanced HCN during all seasons. The highest HCOOH amounts in the southern tropics and mid-latitudes with maximum values of up to 160 ppt above tropical South America and Africa were observed in austral spring (Fig. 5f). [1] During this season enhanced HCOOH coincides well with the HCN plume extending from Brazil over southern Africa and the southern Indian and Pacific Ocean. The good spatial correlation with HCN in these regions is a strong indication for a significant contribution from biomass

burning to upper tropospheric HCOOH during austral spring.

Due to practically year-round biomass burning signatures over Africa, a clear detection of biogenic release of HCOOH in this region is difficult in the seasonal composites presented in Figure 5. The separation between biogenic and biomass burning sources of HCOOH is somewhat easier above tropical South America. Here, local maxima of enhanced HCOOH are visible during austral spring and summer (Fig. 5f,h). Especially the maxima during December to February are not correlated

with HCN, which hints to direct biogenic emission or generation from other biogenic sources as, e.g., isoprene (Keene and Galloway, 1988). Rather clear indication of biogenic HCOOH release in the Amazon region during July and August can be seen in Figures A2 and A3 in the Appendix.

Besides in austral spring, at least slightly enhanced HCOOH amounts were also observed in the latitude band 30°S − 60°S in all other seasons. Differently to the period September to November, these enhancements are not so well correlated with HCN,

which hints to a strong contribution from biogenic sources. Relevant source regions are tropical South America and tropical Africa. Figure S7 of the Supplements shows, how, e.g., during August 2010 air masses from the boundary layer above Brazil are transported into the UT in this latitude band. According to Franco et al. (2020, their Fig. 4) especially the Amazon rain

---

[1]Note that the HCOOH distributions during austral spring and summer are displayed at the altitude of 5 km below the tropopause, because some features of tropical HCOOH become better visible at this level.




forest is a strong source of HCOOH-precursors like isoprene and monoterpenes. Their fluxes strongly increase during austral winter and peak in austral spring around September. At southern high latitudes very low HCOOH amounts were measured during most of the year, with a slight increase up to about 70 pptv during austral spring.

## 4.2    Monthly volume mixing ratios in different latitude bands

Figure 6 shows monthly averaged VMRs observed by MIPAS in 10°-wide latitude bands at 3 km below the tropopause. As already shown in Figure 2, HCN exhibits two maxima of about equal strength, one located in the northern and the other in the southern hemisphere (Fig. 6a). Northern hemispheric HCN peaks in May at mid-latitudes and in August at high latitudes, reflecting the northward progression of biomass burning. The enhanced HCN amounts observed from May to August between 10° and 40°N are caused by the northern African plume and by accumulation of HCN in the AMA starting in June. In
the southern hemispheric tropics and mid-latitudes, strongly elevated HCN amounts were observed between September and December, with maxima during October–November. Thereafter, the HCN amounts continuously decrease during austral winter and spring. At high southern latitudes maximum HCN amounts were measured in December, which is due to the time required for southward transport from the biomass burning regions in the SH. Compared to southern hemispheric lower latitudes, the decrease begins one month later. The lowest HCN VMRs in the northern extra-tropics were observed in December, in the
northern tropics during January–February, in the southern tropics during March–April and at southern mid- to high latitudes during May–June. In the latter period the HCN amounts decrease to about 150 pptv.

At northern mid- and high latitudes the monthly variation of CO (Fig. 6b) exhibits similarities to HCN, but the CO maximum in summer is less distinct. Instead, high CO amounts were observed during most of the year, indicating release by other anthropogenic sources in addition to biomass burning. Enhanced CO was also measured in the northern tropics in spring and
in the northern subtropics in summer. The CO maximum in October–November in the southern tropics is well correlated with HCN, which hints at release by biomass burning. Southward expansion of polluted air masses is visible in slightly enhanced CO at high southern latitudes between September and November. However, there is already a strong decrease between southern mid- and high latitudes in December.

In the northern extra-tropics, strong maxima of $C_2H_2$ and $C_2H_6$ were observed in February–March and minima in summer
at high latitudes and in fall at mid-latitudes (Figs. 6c,d). Due to its shorter lifetime under sunlit conditions, $C_2H_2$ decreases considerably stronger than $C_2H_6$. Ethane also exhibits enhanced VMRs in the northern tropics in spring and in the northern subtropics in summer. The more localized $C_2H_2$ maxima in these regions shown in Fig. 4 are hardly visible here due to zonal averaging. Due to their shorter lifetimes, the $C_2H_6$ and $C_2H_2$ enhancements in southern hemispheric spring are temporally more restricted than those of HCN, with maxima in September–October and a strong decrease in December.

In the northern extra-tropics both PAN and HCOOH exhibit maxima in summer and minima in winter (Figs. 6e,f). This similarity to HCN points to release by biomass burning. However, as mentioned above, other anthropogenic and biogenic sources are possible as well. We will try to get more information on their origin by the correlation analysis in Sect. 4.3. In the representation chosen here, HCOOH enhancements are hardly visible in the northern tropics and subtropics. In the southern tropics and at southern mid-latitudes, maximum PAN amounts were observed in October and November, which is in phase with





the HCN cycle. This is a strong indication for release of PAN by biomass burning. Maximum southern hemispheric HCOOH amounts were measured during September to October. At southern mid-latitudes, HCOOH already increases considerably in August, i.e. one month before HCN. As mentioned above, this points to biogenic release of HCOOH, which in the following months becomes superposed by biomass burning. While enhanced PAN amounts were observed up to 50°S, enhanced HCOOH

extends further southward up to 60-70°S.

## 4.3 Global tracer-tracer correlation and regression analysis

In the following, we discuss the results of correlation and regression analyses between the biomass burning tracer HCN and the other pollutants to obtain a more quantitative assessment of their sources.

### 4.3.1 Monthly correlation coefficients in different latitude bands

Figure 7 shows Pearson correlation coefficients $r$ obtained for monthly composites of single-scan MIPAS data in 5°-wide latitude bands at 3 km (50°S – 50°N) and at 2 km (higher latitudes) below the the tropopause. The lower distance to the tropopause at high latitudes was chosen to ensure a sufficiently high amount of data points for the correlation analysis in these regions.

The pollutants $C_2H_2$ and PAN exhibit the highest correlations with HCN (Fig. 7b,d). In both cases, correlation coefficients

between $r = 0.7$ and $r = 0.85$ were obtained for observations between low southern and northern mid-latitudes during boreal spring and between the equator and southern mid-latitudes during austral spring. This is a strong indication that - like for HCN - at these times biomass burning was the major source of enhanced $C_2H_2$ and PAN in these areas. Beyond that, high PAN-HCN correlations were additionally observed at low latitudes in February, June and during November to December. High $C_2H_2$-HCN correlation coefficients also prevail at northern mid- and high latitudes during boreal summer and early fall,

pointing to release of $C_2H_2$ by biomass burning as well. From October to March the $C_2H_2$ and HCN amounts at northern high latitudes are uncorrelated, confirming the dominant influence of the OH cycle and of biofuel burning on the $C_2H_2$ amounts in these regions during winter. North of 50°N, PAN and HCN amounts are not or only weakly correlated between November and June, indicating that the strong PAN increase in the northern extra-tropics during spring and early summer is not dominated by biomass burning, but has anthropogenic or biogenic sources. This observation agrees with model results of Fischer et al. (2014),

who identify anthropogenic and biogenic activities as major sources of northern extra-tropical PAN during boreal spring and summer (cf. Sect. 4.1.5). The increasingly higher correlation coefficients at northern mid- and high latitudes during July to September hint at growing contributions from boreal fires in late summer.

The correlations of CO, $C_2H_6$ and HCOOH with HCN are generally weaker (Fig. 7a,c,e). However, they are also characterized by maxima at southern tropics to mid-latitudes during austral spring ($r$ between 0.5 and 0.7) and somewhat weaker

maxima in the northern subtropics and mid-latitudes during boreal spring. Moderately high correlations ($r = 0.5 − 0.65$) at northern mid- and high latitudes during boreal summer point to biomass burning as a common source as well. Compared to $C_2H_2$ and PAN, the lower correlations at low latitudes indicate a larger contribution of other anthropogenic or biogenic sources to the enhanced CO, $C_2H_6$ and HCOOH observed there.





### 4.3.2 Correlation coefficients and enhancement ratios in latitude-longitude bins

We now discuss correlation coefficients and enhancement ratios (ERs) in $5° \times 15°$ latitude-longitude bins for selected monthly composites at 3 km ($50°S - 50°N$) and 2 km (higher latitudes) below the tropopause. As an example of the underlying data, scatter plots of $C_2H_2$ versus HCN for a bin above southern Africa are presented in Fig. S9 of the Supplement.

Enhancement ratios are used to gain information on the type of fuel burned by comparison of remotely measured data with emission ratios determined directly at the source. The ER of species X is defined as the difference $\Delta X$ between its mixing ratio in a certain plume and its mixing ratio in background air, divided by the difference $\Delta Y$ of a widely non-reactive co-emitted tracer like CO or $CO_2$ (Lefer et al., 1994; Hobbs et al., 2003; Akagi et al., 2011). In contrast, emission ratios are measured directly at the sources. They can be derived from emission factors (emitted compound among biomass burned in g kg$^{-1}$)

determined for various trace species by burning under laboratory conditions or by measurements close to open or domestic fires. Measurements of emission factors for burning of various types of vegetation have, e.g., been performed by Bertschi et al. (2003) or Yokelson et al. (2003, 2008, 2009) and by other authors referenced therein. Comprehensive compilations have been given by Andreae and Merlet (2001), Akagi (2011) and by Andreae (2019). In comparing ERs with emission ratios it has to be considered that both can differ from each other due to plume aging (faster depletion of shorter-lived species).

We calculated monthly $\Delta HCN/\Delta CO$ and $\Delta X/\Delta HCN$ (X = $C_2H_2$, $C_2H_6$, PAN or HCOOH) enhancement ratios by linear Deming regression analysis (Deming, 1943). This method of regression analysis, which is equivalent to the maximum likelihood estimation of the slope of the regression, was chosen because of uncertainties of different magnitude both in HCN and in the other pollutants, and implemented as described in Mikkonen et al. (2019). In case of equal error variances for both variables, Deming regression simplifies to orthogonal regression. The $\lambda$-values required for Deming regression were determined

from the ESDs given in Tab. 2 as $\lambda = ESD_y^2/ESD_x^2$. After averaging over the two height levels in Tab. 2, the following global $\lambda$-values were applied: $\Delta HCN/\Delta CO$: $4.42 \times 10^{-6}$, $\Delta C_2H_2/\Delta HCN$: 0.35, $\Delta C_2H_6/\Delta HCN$: 20.01, $\Delta PAN/\Delta HCN$: 0.63 and $\Delta HCOOH/\Delta HCN$: 0.75.

    We compare MIPAS ERs with values obtained from ACE-FTS observations of individual plumes and with emission ratios for burning of different types of vegetation (Tab. 3), derived from emission factors given in the most recent compilation of Andreae

(2019). The values in parentheses in Table 3 show that the standard deviation or natural variation of the emission factors and consequently of the emission ratios generally is larger than 50% or even 100% for each type of fuel. These variations are mostly larger than the differences between the emission ratios for the fire types to be considered in a certain global region. One exception is the emission ratio for peat fires, which compared to other types of biomass burning is very high for $\Delta HCN/\Delta CO$ and very low for all other emission ratios contained in Table 3. The reason is the high HCN emission factor for this type

of smoldering combustion. Another exception is the high $\Delta C_2H_2/\Delta HCN$ emission factor for biofuel burning. Further, the $\Delta C_2H_6/\Delta HCN$ and $\Delta HCOOH/\Delta HCN$ emission factors for savanna burning are considerably lower than those of the other types of burning (tropical forest, agricultural residues, biofuel) in tropical or subtropical regions.

    Because of the different lifetimes of the pollutants, the calculated $\Delta HCN/\Delta CO$ enhancement ratios might potentially be higher than the respective emission ratios observed directly at the fires, while the $\Delta X/\Delta HCN$ enhancement ratios might be





lower. On the other hand, $\Delta HCOOH/\Delta HCN$ and $\Delta PAN/\Delta HCN$ might also become higher than the emission ratios due to post emission processing. Further, the different vertical resolutions (VRs) of the correlated gases (cf. Tab. 2) have to be considered. Fortunately, the VRs of HCN and CO are rather similar at 2–3 km below the tropopause, i.e. at $\sim$ 14 km in the tropics and at $\sim$ 8 km at higher latitudes. Except for tropical $C_2H_6$, the VRs of the other pollutants are clearly different from the one of

HCN. However, degradation of, e.g., the $C_2H_2$ profiles to the height resolution of HCN on average resulted in 10% lower $C_2H_2$ VMRs, which would lead to 10% lower ERs only. Due to these caveats, comparison of the enhancement ratios presented in the following with emission ratios has to be performed with caution. But the consideration of ERs in addition to the correlation coefficients has a synergistic effect: High correlations with HCN in combination with ERs in the range of the listed emission ratios both indicate biomass burning. On the other hand, enhancement ratios far out of the range of possible emission ratios

point to other sources of the other pollutants even in the case of a high correlation with HCN.

For statistical reasons we only show correlation coefficients and ERs for bins with at least 10 data pairs in the following plots. To restrict the presentation to bins with enhanced VMRs of the respective pollutant, further filter criteria are mean bin values of 80 ppbv for CO, 60 pptv for $C_2H_2$, 300 pptv for $C_2H_6$, 100 pptv for PAN and 60 pptv for HCOOH. For comparisons with the emission ratios in Table 3 and enhancement ratios of ACE-FTS we calculated mean ERs from all bins with correlation

coefficients $r > 0.6$ ($> 0.55$ for CO) in eight different global regions (Figure 8). For selected months, these ERs are shown in Table 4.

**HCN versus CO**

Figure 9 shows global distributions of the correlation coefficients (left column) and enhancement ratios (right column) obtained from the correlation analysis between HCN and CO for February, April, July and October (top to bottom).

During February, moderately strong CO-HCN correlations (around $r=0.6$) were observed above North-East Africa and the Middle East, the southern tropical Atlantic and the eastern North-Pacific around 40°N, indicating potential common release of HCN and CO by biomass burning into these air masses. The respective mean $\Delta HCN/\Delta CO$ enhancement ratios, calculated for bins with correlation coefficients $r > 0.55$, are $(0.18\pm0.03)$% for Northern Africa, $(0.18\pm0.04)$% for Southern Africa and $(0.25\pm0.06)$% for the Northern Pacific (see Tab. 4). The values around Africa are considerably lower than the emission ratio

of 0.66% for savanna and grassland fires listed in Tab. 3, which according to, e.g., Yokelson et al. (2003) and van der Werf et al. (2010, 2017) dominate the fire carbon emissions in Africa. They are also lower than the emission ratios of the other types of fuel possibly burned in these regions, namely tropical forest (0.44) or agricultural residues (0.57), listed in Tab. 3. The same applies for the observations above the East Pacific. This hints to additional anthropogenic sources of the observed CO amounts, leading to lower enhancement ratios.

During April there are strong HCN-CO correlations of up to $r = 0.8$ above the Middle-East and in the latitude band 20-50°N between the North Pacific and the North Atlantic, which strongly indicate common release by biomass burning, and moderately high correlations around $r = 0.5$ above South Africa. The mean enhancement ratios above North and South Africa have hardly changed and amount to $(0.15\pm0.03)$% and $(0.18\pm0.05)$%, respectively. The mean ER above the Northern Pacific, the United





States and the Northern Atlantic is (0.29±0.06)%, which is somewhat closer to the emission ratios in Tab. 3. Further, this value is in very good agreement with the enhancement ratio of 0.26% (all data) or 0.34% (filtered for low $\Delta C_2Cl_4$) for measurements in the free troposphere above the Northern Pacific during spring 2001, reported by Singh et al. (2003).

    The month of July is characterized by high correlations above East Siberia and Alaska, but also scattered above Canada and the Arctics, pointing to common release of HCN and CO by boreal fires. The mean enhancement ratio for bins with $r > 0.55$ is (0.29±0.11)% for North America and (0.28±0.11)% for Siberia, which corresponds very well with the values for boreal fires given in Tereszchuk et al. (2013b). These authors obtained ERs of (0.20±0.09)% – (0.25±0.12)% for North American and of (0.26±0.02)% – (0.30±0.11)% for Siberian biomass burning plumes aged between 0 and 7 days from ACE-FTS measurements during the BORTAS campaign from 12 July to 3 August 2011. Based on FTIR-measurements of ground-based column amounts at Eureka and Thule during 2008–2012, Viatte et al. (2015) determined average ΔHCN/ΔCO ERs of (0.33±0.09)% and (0.43±0.25)%, respectively, for mostly boreal forest fires. Both the ERs determined from MIPAS and ACE-FTS measurements are lower than the emission ratio of (0.45±0.31)% for boreal forest fires derived from the values given in Andreae (2019) (Tab. 3), but well within its standard deviation. On the other hand they are higher than the value of 0.15% derived from the emission factors for extratropical forest given in Andreae and Merlet (2001). In any case, as already stated by Tereschuk et al. (2011), these observations indicate that the HCN/CO emission ratio of (1.24±0.80)% derived from the emission factors in Akagi et al. (2011) is too high. The North American and Siberian ERs for August also shown in Tab. 4 are higher, namely (0.39±0.13)% and (0.42±0.21)%, which agree very well with the values given by Andreae (2019). Like in February and April, there is an area of moderately high correlations above southern Africa and the southern tropical Atlantic with a mean ER of (0.22±0.02)%. Again, this ER is lower than to be expected from the regional types of biomass burning.

    The October data show a large area of high correlations up to $r = 0.85$ extending from tropical South America to Southern Africa and over wide regions in the southern tropics to mid-latitudes, which strongly points to biomass burning as common source of both pollutants. This is confirmed by the mean ERs of (0.50±0.09)% and (0.43±0.13)% obtained for South Africa and South America, respectively, which are in good agreement with the emission ratios of 0.66% and 0.44% for savanna and tropical forest fires in Tab. 3. While the major contribution to pollution in southern Africa is attributed to savanna or woodland fires, pollution in South America approximately in equal parts results from savanna/shrubland fires and from tropical forest fires (van der Werf, 2017). There is also good agreement with the enhancement ratios given by Tereszchuk et al. (2011, their Tab. 1) for ACE-FTS measurements in plumes released from the Congo and the Amazon region, which are (0.55±0.02)% and (0.36±0.01)%, respectively. While the ΔHCN/ΔCO ERs for savanna and tropical forest fires derived from the emission factors given in Akagi et al. (2011) are nearly identical to those in Tab. 3 derived from the emission factors in Andreae (2019), the respective values on the basis of the compilation of Andreae and Merlet (2001) are (0.04±0.02)% and 0.15, which in comparison to the observed ERs seem to be clearly too small.

    The highest October enhancement ratios of (0.65±0.13)% and of (0.68±0.18)% for bins with $r > 0.55$ were obtained for measurements above the tropical Indian Ocean/Indonesia (IOAUS), and above southern hemispheric midlatitudes (SHMIDL), respectively. Figure 9h shows that the high ERs at southern mid-latitudes are focused to the eastern Indian Ocean, which is adjacent to the Indonesian/Australian region. The value for the region IOAUS for November (not shown) is even higher, namely



(0.85±0.24)%. In so-called El Niño years this area is strongly polluted by biomass burning in Indonesia. Biomass burning in Indonesia is characterized by a high percentage of peatland fires, which nearly completely burn under smoldering conditions (Akagi et al., 2011). For this type of fire Andreae (2019) specifies large HCN emission factors, which lead to a ΔHCN/ΔCO emission ratio of 1.75% (Tab. 3). Thus we conclude that a significant portion of peatland burning is the reason for the high

ΔHCN/ΔCO enhancement ratios observed above the eastern Indian Ocean. The measured ERs for the October composite are still considerably lower than the emission ratio derived from the Andreae-data, because biomass burning in Indonesia is significantly intensified in El Niño years only. During the MIPAS measurement period there were two strong (2002 and 2006) and one minor (2009) El Niño years. In all other years, enhanced HCN and CO amounts over the eastern Indian Ocean presumably were dominated by African or Asian fires with lower HCN emission factors.

As can be seen in Fig. S11 of the Supplement, most of the spatial patterns of high correlation coefficients visible in the composite plots presented here also appear in the single-year distribution of 2008. Further, the single-month ERs are mostly of the same magnitude as the monthly composites.

**$C_2H_2$ versus HCN**

High correlations between $C_2H_2$ and HCN occur in even larger regions than for HCN and CO and they are generally stronger,

often reaching values of $r$ = 0.8 or above (Fig. 10, left column). Again, the focus is above North-East Africa, the southern tropical Atlantic and the North Pacific in February and April. In August, there are high correlations at northern mid- and high latitudes, above tropical Africa, in the AMA region and above the southern Indian Ocean, and in October very high correlations above the southern tropical Atlantic, South Africa, the Indian Ocean and Australia. All these features strongly point to biomass burning as common source of $C_2H_2$ and HCN.

The mean $\Delta C_2H_2/\Delta HCN$ enhancement ratio obtained for February for the region above northern tropical Africa and the Middle East is 0.80±0.17 (Fig. 10b). This value - also after a 10%-subtraction as discussed above - is in good agreement with the emission ratio for the dominant burning process in northern tropical Africa, namely savanna or grassland fires (0.73). However, the emission for tropical forest fires (0.83) or burning of agricultural residues (0.67) are not much different. The mean ER of 0.95±0.19 for southern tropical Africa is somewhat higher. A possible reason is additional release of $C_2H_2$ from biofuel

burning, e.g. cooking fires or domestic heating, with a higher emission ratio of 1.81 (Tab. 3). The average ER for the region with high correlations around 40°N above the Pacific and North America is 1.83±0.37. Biofuel burning might also come into consideration for this high value, but due to the spatial non-uniformity of the correlation coefficients and ERs in this region we also assume mixing with $C_2H_2$-rich air masses from further north as possible contribution.

  In April, the area of high $C_2H_2$-HCN correlations above Africa has increased and expanded over South Asia and the Chinese

Sea. Further, there is a uniform region of very high correlations around 30°N, covering the North Pacific, the United States and the North Atlantic. Compared to February, the mean enhancement ratio above North Africa and South Asia has decreased to 0.61±0.10 and exhibits a general decline towards the east, which is an indication of more aged pollution. The ER obtained above southern Africa has not changed much and is 0.94±0.31. The mean ER in the band with high correlations around



30°N (NHSUB) is 1.00±0.31. This value is clearly lower than the ER determined for February and only somewhat above the emission ratios of potential sources, namely tropical forest fires, burning of agricultural residues and savanna fires. This shows the progressive upper tropospheric pollution by biomass burning in tropical East Asia and additionally in North Africa during spring.

In August, the main burning activity in Africa has shifted southward across the Equator, and there are very high $C_2H_2$-HCN correlations above southern tropical Africa, the tropical Atlantic and in a band around 45°S in the eastern hemisphere. High to very high correlations were also observed at northern mid- and high latitudes and in the AMA region. All these features indicate common release of HCN and $C_2H_2$ by biomass burning. As confirmation, the ERs obtained for South Africa and South America are 0.77±0.17 and 0.66±0.07, which agree very well with the emission ratios for savanna and tropical forest fires. The

area above North Africa and the Middle-east (NAF), covered by the western part of the AMA, exhibits an ER of 1.06±0.23, pointing to contributions from biofuel burning. The mean ER for the bins of high correlation above North America and Siberia is 0.64±0.23 and 0.57±0.30, which is in good agreement with the emission ratio for boreal forest fires (0.55±0.40).

    In October, high correlations with HCN prevail over large areas in the southern hemisphere, which are even stronger than those between HCN and CO. The mean ERs for South Africa and South America are 0.32±0.04 and 0.33±0.05, respectively,

which is much lower than the values of August and the emission ratios coming into question. Presumably this is an effect of plume aging, in this case stronger accumulation of HCN due to its relatively long lifetime. Compared to South America and South Africa, the mean ER above the Indian Ocean and Australia is even lower, namely 0.25±0.09, which also points to a contribution from peat fires.

    As an alternative approach, Fig. S10 of the Supplement shows scatter plots of September and October composites of $C_2H_2$

versus HCN for the whole southern Africa. The respective ER determined for October is 0.31, which agrees very well with the bin-average given above. In Fig. S12 we present monthly distributions of $C_2H_2$-HCN correlation coefficients and ERs for the single year 2003. The spatial patterns of high $C_2H_2$-HCN correlation coefficients are very similar to those of the monthly composites discussed above. Further, the single-month ERs generally agree well with those for monthly composites.

## $C_2H_6$ versus HCN

In most regions and months the $C_2H_6$-HCN correlations (Fig. 11, left column) look very similar to those between CO and HCN, reflecting the same regional patterns of biomass burning. Like for HCH versus CO, the correlation coefficients are lower than those between $C_2H_2$ and HCN, indicating a weaker connection. But it has to be considered that a certain part of the weaker correlations is due to the up to two times larger relative retrieval error of $C_2H_6$ as compared to $C_2H_2$.

    The distribution of $\Delta C_2H_6/\Delta HCN$ enhancement ratios is characterized by values considerably higher than 3 in nearly

the whole northern hemispheric extra-tropics during February, April and October (Fig. 11, right column). These ERs are far above the emission ratios for biomass burning (Tab. 3), and $C_2H_6$ and HCN are mostly uncorrelated or only moderately well correlated in this region as well. Both findings point to other sources of enhanced $C_2H_6$ at northern mid- and high latitudes during these months, namely accumulation in boreal winter due to the minimum in the OH cycle and anthropogenic activities.



During August, very high ERs above 3 also cover a region extending from the Gulf of Mexico and the central United States above the northern Atlantic towards Europe as well as from the tropical East Pacific to north-western South America. In these regions $C_2H_6$ is also not correlated with HCN. Thus, both the high ERs and the missing correlation indicate that these $C_2H_6$ enhancements are not caused by biomass burning but by losses of ethane in the oil and gas production and transport (cf. Appendix). During April, i.e. before the onset of the North American monsoon, high ERs were also observed, but above northern South America and the tropical East Pacific only.

The enhancement ratios in the regions with moderately high (February) and high correlation coefficients (April) are generally much lower. During February and April, mean values for bins with $r > 0.6$ of 1.80±0.30 and of 1.73±0.47 were calculated for North Africa and the Middle-East, respectively. These values are clearly higher than the emission ratio for savanna or grassland fires (0.86), but in good agreement with the emission ratios for tropical forest fires (1.80) or burning of agricultural residues (1.69). This either implies a dominance of tropical forest burning instead of savanna fires or additional anthropogenic sources of the enhanced $C_2H_6$. The ERs in the northern hemispheric subtropics are considerably higher, which points to additional sources and to mixing with air-masses from further north. But like for $C_2H_2$, the considerable decrease from 5.87±2.85 in February to 2.94±1.96 in April indicates the growing contribution of biomass burning.

In August, the mean ERs for the bins with high correlation coefficients above North America and Siberia are 2.37±0.72 and 2.37±0.83, respectively, which is larger than the emission ratio for boreal fires (1.65). Again, this points to a combination of boreal forest fires and other sources leading to the enhanced $C_2H_6$ in these regions. The mean ERs in bins with high correlations above South Africa is 1.04±0.19, which is somewhat above the emission ratio for savanna fires (0.86), indicating savanna burning and possibly a certain contribution of tropical forest fires.

The mean ERs for the large area of high southern hemispheric $C_2H_6$–HCN correlations in October are 0.86±0.09 above South Africa and 0.93±0.11 above South America, which fit very well to the emission ratio for savanna and grassland burning. This is the dominant biomass burning process in Southern Africa during this time of the year, when the main fire region has moved further southward. Probably due to the longer lifetime of $C_2H_6$, the October ER for Southern Africa contrary to $\Delta C_2H_2/\Delta HCN$ does not yet reflect plume aging. In South America, savanna burning and tropical forest fires have approximately equal contributions to the annual carbon emisssions (van der Werf et al., 2017, their Fig. 9), but this is not reflected in the upper tropospheric ERs obtained from the MIPAS measurements. The mean ER above the Indian Ocean and Australia is somewhat lower (0.74±0.22), which again points to conributions from peat fires.

**PAN versus HCN**

The PAN-HCN correlations (Fig. 12, left column) are comparably strong as those between $C_2H_2$ and HCN. During February, they are very high above Africa, the Middle-East and the southern tropical Atlantic as well as in a smaller region above the North Pacific. The respective mean ERs are 1.78±0.28 above North Africa and the Middle-East, 1.69±0.26 above South Africa, 1.62±0.33 above South America and 1.22±0.22 in the latitude band 20°N–45°N. Probably due to its formation as secondary product of biomass burning there is hardly information on emission factors of PAN in the literature. From the ERs



given in Tereszchuk et al. (2013b) for ACE-FTS measurements in boreal fire plumes, we derived ΔPAN/ΔHCN emission ratios covering the range of 0.9 to 2.75, most of them lying between 1 and 2.

In April, the region of high correlations has considerably expanded and nearly covers the whole latitude band from 10°N to 40°N as well as the area between the tropical southern Atlantic, the southern Indian Ocean and Australia. The respective ERs are 1.62±0.28 (NHSUB), 1.27±0.20 (SAM), 2.12±0.77 (SAF) and 1.46±0.32 (IOAUS), which is in the same range as in February. At northern mid- and high latitudes PAN is generally not correlated with HCN and the ERs are as high as 4 or more. Both findings indicate that the NH PAN increase in boreal spring rather has biogenic sources.

In July, the high correlations in the band 10°N to 40°N are restricted to the region from the Middle East to the mid-Pacific. The mean ER above North Africa and the Middle-East, covering the western part of the AMA, is 1.93±0.54 while it has increased to 2.66±0.91 in the region NHSUB. The band is interrupted by an area of low correlations, extending from the Gulf of Mexico above the tropical Atlantic to Europe. Low correlations were also observed over Europe, Siberia, Northern Canada and the Arctics. All these regions are characterized by very high ERs of more than 4. Both the low correlations and the high ERs hint at major contributions from anthropogenic or biogenic sources for the PAN enhancements as outlined in the Appendix. As shown in Fig. 7, the northern extra-tropical PAN-HCN correlations in August and September are higher, which indicates PAN release from biomass burning during this time. Another region with high correlations in July is southern Africa and the adjacent southern Atlantic. The mean ER for this region, dominated by pollution from biomass burning, is 1.16±0.17, which is lower than the northern hemispheric values.

Like for the other pollutants, there is a large region of high correlations in the southern hemisphere in October, extending from tropical South America over the southern Atlanic to southern Africa and continuing in a band at southern mid-latitudes. High correlations were also observed in adjacent regions above North East Africa and the East-Pacific. The correlations are comparably strong like those for $C_2H_2$ versus HCN. The mean ERs are 1.09±0.27 above South America, 1.00±0.28 above South Africa and 0.61±0.20 above Indonesia and Australia. Again, the low value in the region around Indonesia and Australia (IOAUS) might also be caused by contributions of peat fires. The somewhat higher ERs above North Africa and eastward of tropical South America point to additional or different pollution processes.

## HCOOH versus HCN

In February, there are only a few bins with mean HCOOH VMRs larger than 60 pptv. A contiguous area of high correlations is situated above North-East Africa. The respective mean enhancement ratio for bins with $r > 0.6$ is 0.74 ±0.26, which is clearly above the emission ratio for savanna burning (0.28), but close to the values for tropical forest fires (0.65) and for burning of agricultural residues (0.78). Like for $\Delta C_2H_6/\Delta HCN$ this either implies observation of pollution by tropical forest fires or burning of agricultural residues instead of savanna fires, or biogenic contributions to the enhanced HCOOH. The other regions around 45°N, South America and the southern tropical Atlantic mostly exhibit moderately strong correlations only, and a larger variation both in $r$ as well as in the ERs, giving no clear picture.



In April, the area of high correlations above North-East Africa has somewhat increased. The mean ER in this region is 0.57±0.15, which corresponds well with the emission ratios for tropical forest fires. But, compared to February, it can also be an indication for aged pollution. Similar as for the other pollutants, very high correlations were observed in the latitude band 20°N–40°N above the Pacific, the United States and the North Atlantic. The mean ER in this area is 0.77±0.23, which fits well to the emission ratios for tropical or temperate forest fires and for burning of agricultural residues. There are only low to moderate correlations north of 40°N, indicating that the northern hemisperic HCOOH increase in spring (cf. Fig. 5) is not caused by biomass burning but rather by oxidation of biogenic precursors.

During August, most of the bins in the northern hemispheric extra-tropics exhibit moderately strong to strong correlations with HCN, with a focus above Siberia and northern North America. The mean ER for bins with $r > 0.6$ is 1.04±0.60 for North America and 1.23±0.46 for Siberia, which is in good agreement with the emission ratio of 1.15 for boreal forest fires. Thus, both the high correlations as well as the ERs indicate release of HCOOH by boreal fires. But regions with lower correlations and high ERs, especially above the North Eastern United States, point to biogenic sources as well (cf. Appendix, Figs. A2, A3). Strong correlations were also observed above southern hemispheric tropical Africa and in a band between 30°S and 50°S, extending from the South Atlantic eastward as far as New Zealand. The mean ER above southern Africa is 0.80±0.64. Restricted to southern tropical Africa it is even lower, namely 0.49±0.08, which is in good agreement with the emission ratio for tropical forest fires and savanna burning. In comparison, the correlation above South America is lower and the mean ER is 1.21±0.26, which is clearly above the emission ratios for biomass burning, indicating biogenic release (cf. Fig A3). The ERs in the southern hemispheric band at 40°S are much higher, namely around 1.5 above the southern Atlantic and up to 2.5 or larger south of Australia and above the Southern Pacific. Despite of the high correlations with HCN south of Africa and south of Australia, the high ERs rather indicate a strong portion of biogenic sources of the enhanced HCOOH in this latitude band during this time of the year. As shown in Fig. A3, a clear region of biogenic sources is tropical South America, but sources in Central/Western Africa can contribute as well.

In October there are strong correlations in the area between South America and South Africa and somewhat lower ones in the region extending eastward to Australia. The mean ERs above South America and South Africa are 0.52±0.20 and 0.38±0.09, respectively. The first value is in good agreement with the emission ratio for tropical forest fires and the second in-between the latter and the value for savanna burning. Thus both the high correlations as well as the ERs indicate that biomass burning is a major source of the enhanced HCOOH observed in the southern tropics and subtropics during this period. The correlations in the southern mid-latitude band are not as high and exhibit a larger variation. The ERs above the band segment covering the southern Atlantic and the southern Indian Ocean are nearly as low as in the southern tropics, leading to a mean ER of 0.71±0.26. This indicates that in October the dominant source of the enhanced HCOOH in this region is also biomass burning. However, especially above the South Pacific there are still bins with ERs between 1 and more than 2.5, which point to biogenic sources or secondary HCOOH formation as well.





## 5 Summary and conclusions

We compared the seasonal distributions of upper tropospheric HCN, CO, $C_2H_2$, $C_2H_6$, PAN and HCOOH measured by MIPAS between July 2002 and April 2012 to distinguish between different sources of upper tropospheric pollution. While enhanced HCN is regarded as nearly unambiguous tracer of biomass burning, the other pollutants have anthropogenic or biogenic sources as well.

At northern mid- and high latitudes, maximum HCN, CO, PAN and HCOOH amounts were measured during spring and/or summer and minima during winter. On the contrary, maximum northern extra-tropical $C_2H_2$ and $C_2H_6$ amounts were observed during winter and spring and minimum values during summer and fall. The reason is the minimum of their dominant reactant of $C_2H_2$ and $C_2H_6$, the OH radical, as well as enhanced biofuel and fossil fuel burning during winter. In the tropics and subtropics, enhanced amounts of all pollutants were observed during each season, especially widespread in the southern hemisphere during austral spring. Due to its long lifetime, enhanced HCN was even measured at high southern latitudes during austral summer. Another characteristic feature observed is trans-Atlantic transport of $C_2H_6$ and PAN in boreal summer. This transport pattern is not accompanied by enhanced HCN, which points to $C_2H_6$ release from fossil fuel production and transport around the Gulf of Mexico and subsequent formation of PAN. Simultaneous observation of enhanced HCOOH indicates biogenic sources in the Eastern United States as well. Biogenic sources of HCOOH can also be identified in the Amazon region and, with certain limitations, in tropical Africa. Further measurements are accumulation of pollutants in the Asian Monsoon Anticyclone during boreal summer and enhanced $C_2H_2$ over South-East Asia in boreal winter.

To quantify the assumptions on the sources of the other pollutants, we additionally performed correlation analyses with HCN in latitude-longitude bins covering the whole upper troposphere. These analyses mostly confirmed the conclusions obtained from comparison of the VMR distributions. E.g., in the area of the southern hemispheric biomass burning plume the other pollutants are very good correlated with HCN. High correlation of $C_2H_2$ and PAN with HCN in the tropics during all other seasons as well generally points to biomass burning as important source of these pollutants, while the weaker correlation of CO, $C_2H_6$ and HCOOH with HCN hints at additional anthropogenic and biogenic sources. The analysis also showed that the VMR enhancements of CO, PAN and HCOOH in large regions north of 50° during boreal spring and early summer are rather uncorrelated or only moderately well correlated with HCN, indicating other anthropogenic and biogenic sources. In August and September, during and after the peak of the boreal fire season the respective correlations become better, pointing to release of these species by boreal fires as well.

Although the $\Delta$HCN/$\Delta$CO enhancement ratios presented here are monthly composites, there is good agreement with ACE-FTS values derived from individual plumes in different global regions. Further, the enhancement ratios determined for mid- and high latitudes in July are in-between the emission ratios derived from the emission factors for extra-tropical or boreal forest fires given in Andreae and Merlet (2001) and in Andreae (2019), but a factor of 3–5 lower than the emission ratios corresponding to the values listed in Akagi et al. (2011). The $\Delta$HCN/$\Delta$CO ERs obtained for the plume above northern tropical Africa and the Middle-East during boreal spring are clearly below the respective emission ratios for biomass burning, indicating additional sources of the observed CO. In the southern hemispheric plume the observed ERs agree well with the emission ratios



for tropical forest fires and savanna burning derived from the emission factore given in Andreae (2019). They also match to the values given in Akagi et al. (2011), but are significantly higher than the emission ratios corresponding to the values for savanna burning and for tropical forest fires of Andreae and Merlet (2001). The highest $\Delta HCN/\Delta CO$ enhancement ratios, observed above the tropical Indian Ocean and northern Australia, are most probably caused by intensive peat fires in Indonesia during

the El Niño years 2002/2003 and 2006/2007.

Although $C_2H_2$ and HCOOH have a considerably shorter atmospheric lifetime than HCN, their enhancement ratios to HCN obtained for northern mid- and high latitudes during August agree well with the emission ratios for boreal fires. The $\Delta C_2H_2/\Delta HCN$ ERs obtained for the pollution above North-East Africa and the Middle-East in boreal spring fit well to the emission ratio for the major local burning process, namely savanna fires, but also to those for burning of tropical forest

and agricultural residuals. The respective $\Delta HCOOH/\Delta HCN$ ERs are clearly higher than the emission ratio for savanna fires and indicate tropical forest fires, burning of agricultural residuals or - considering the dominance attributed to savanna burning - additional biogenic release of HCOOH. The $\Delta C_2H_6/\Delta HCN$ ERs at northern mid- and high latitudes are higher than the emission ratios, indicating additional sources of $C_2H_6$. The respective ERs obtained for pollution above North-East Africa also hint to tropical forest fires rather than savanna burning or alternatively to additional anthropogenic sources. For

southern hemispheric biomass burning, the $\Delta C_2H_2/\Delta HCN$ ERs agree well with the emission ratios for savanna and tropical forest burning during August, while the October ERs indicate aged pollution. The SH $\Delta C_2H_6/\Delta HCN$ ERs are in good agreement with the emission ratios for savanna fires in August as well as in October. Especially above the Amazon region, the $\Delta HCOOH/\Delta HCN$ ERs for August are considerably higher than the emission ratios for savanna or tropical forest fires, indicating biogenic release of HCOOH. In October they fit better to these fire types, which hints to a large impact of biomass

burning to enhanced southern hemispheric HCOOH during this time of the year.

**Appendix: Global distributions at 10–11 km altitude and at 5 km below the tropopause**

Here we present regions of enhanced upper tropospheric pollution, which become better visible at the geometrical altitudes of

10 and 11 km and at 5 km below the tropopause.

**South Asian biofuel burning**

Especially in boreal winter, enhanced $C_2H_2$ VMRs, accompanied by low HCN amounts, were observed above South and

South-East Asia around the altitude of 11 km (Fig. A1). At this height, one $C_2H_2$ plume is situated above Central and North-East Africa and another one above South and South-East Asia (left). Eastward outflow from the Asian plume leads to enhanced $C_2H_2$ above the subtropical northern Pacific, above Central America, the United States and even the northern Atlantic. There is also enhanced HCN over Africa, but only background HCN over South-East Asia and rather depleted HCN above the subtropical Pacific and Central America (right). This indicates that the source of the South-East Asian $C_2H_2$ plume

is not biomass burning but rather biofuel or fossil fuel combustion. Indeed, according to Streets et al. (2003) biofuel burning is



the dominant source of $C_2H_2$ released in South- and South-East Asia.

**Transatlantic transport in boreal summer**

Figure A2 shows the global distribution of $C_2H_6$ and of other pollutants at 10 km altitude for the July composite. Air masses containing enhanced $C_2H_6$ amounts of up to 900 pptv, apparently originating from the Gulf of Mexico and the southeastern United States, are transported over the northern Atlantic towards Europe. Another region of enhanced $C_2H_6$ is the Pacific southwest of Central America. Transport of pollution into the upper troposphere above Central America is favoured by strong convection during summer. The southern U.S. is characterized by very strong convection during the North American monsoon as well (Randel et al., 2012, and references therein). The $C_2H_6$ plume extending from the Gulf region towards Europe is not correlated with $C_2H_2$ or HCN (Fig. A2b, c). This indicates that the enhanced $C_2H_6$ originates from losses during production and transport of oil and gas, e.g., by inproperly operated gas flaring, in the Gulf region. According to Elvidge et al. (2016) the world-wide largest number of flare sites is located in the United States, with a focus near the Gulf of Mexico. A considerable number of further flare sites is in Mexico and in northern South America. Another source is considerably increased fracking in the United States during the late MIPAS measurement period (Franco et al., 2016; Helmig et al., 2016). There is also no correlation with HCN and just a weak correlation with $C_2H_2$ above the Pacific southwest of Central America.

The spatially well correlated plume of enhanced PAN extending from the Gulf of Mexico towards Europe (Fig. A2d) hints at generation of PAN as secondary product of the hydrocarbons released by the oil- and gas industry. However, the nearly congruent plume of HCOOH (Fig. A2e) points to strong biogenic activity as well and thus to possible release of PAN precursors from biogenic sources. A correlation analysis for the pollutants in the plume extending above the northern Atlantic (30-45°N, 30-90°W) for the month of August results in Pearson correlation coefficients of $r=0.50$ for HCOOH versus $C_2H_6$, of $r=0.69$ for PAN versus $C_2H_6$ and of $r=0.65$ for PAN versus HCOOH. This indicates a comparably weak interrelation between HCOOH and $C_2H_6$, but possible industrial as well as biogenic sources of the observed PAN. Biogenic release of the PAN-precursor acetone from the south-eastern United States was observed by Franco et al. (2019) from analysis of data of the Infrared Atmospheric Sounding Interferometer (IASI) on the Metop-A satellite. Measurements of the Ozone Monitoring Instrument (OMI) (http://www.temis.nl/airpollution) show, that a possible source of the $NO_x$ amounts required for PAN formation is anthropogenic pollution in the eastern United States. However, lightning activity can also play a major role. According to Hudman et al. (2007), lightning is the dominant source of the upper tropospheric $NO_x$ observed during summer above the southeastern United States. Compared to the large plume above the northern Atlantic, there is only a weak and spatially very restricted PAN enhancement over Central America. This might be due to lower amounts of $NO_x$ in this region as indicated by OMI measurements.

**Global HCOOH and HCN during August**



In Figure A3 we show global distributions of the August composites of HCOOH and of HCN at 5 km below the tropopause. The HCOOH distribution (left) exhibits one tropical source in the Amazon region and another one in Western/Central Africa. The enhanced HCOOH amounts above tropical South America obviously originate from biogenic sources, since only background HCN amounts were observed there (right). The elevated HCOOH above Western Africa is accompanied by a local

HCN maximum, which however is centered somewhat further south. This hints to biogenic sources of the HCOOH observed in this region but to release by biomass burning as well. While the HCOOH enhancement at southern hemispheric mid-latitudes is as strong as in the tropical source regions, HCN is only moderately enhanced in this region. This indicates that the HCOOH amounts observed at southern mid-latitudes during August are also dominated by biogenic sources in tropical South America and potentially in Western Africa as well.

Like already shown in Figure A2, there is another source of enhanced HCOOH in Central and Eastern North America with outflow over the northern Atlantic. Since these HCOOH enhancements are not correlated with HCN, they seem to have also have biogenic sources. For the HCOOH enhancements above Northern Europe, Siberia and the Canadian Arctics such a clear separation between biogenic sources and biomass burning can not be performed.

*Data availability.* MIPAS HCN, CO, $C_2H_2$, $C_2H_6$, PAN and HCOOH data can be downloaded from the IMK data server (https://www.imk-asf.kit.edu/english/308.php).

*Supplement.* The supplement related to this article is available online at ....

*Author contributions.* NG developed the retrieval setups for HCN, $C_2H_2$, $C_2H_6$, PAN and HCOOH and wrote the paper. GPS and TvC gave helpful comments in interpretation of the global distributions discussed in the paper. BF developed the retrieval setup for CO. SK and AL performed the operational retrievals.

*Competing interests.* At least one of the (co-)authors is a member of the editorial board of Atmospheric Chemistry and Physics. The authors
have no other competing interests to declare.

*Acknowledgements.* The authors like to thank the European Space Agency for giving access to MIPAS level-1 data. Meteorological analysis data have been provided by ECMWF. We acknowledge support by the Deutsche Forschungsgemeinschaft and Open Access Publishing Fund of the Karlsruhe Institute of Technology. MIPAS level-2 data analysis at IMK has been supported by BmWi and BmBF under contract
50EE1547 (SEREMISA). B. Funke acknowledges support by the Spanish MCINN (ESP2017-87143-R and PID2019-110889RB-I00) and EC FEDER funds. The authors gratefully acknowledge the NOAA Air Resources Laboratory (ARL) for the provision of the HYSPLIT transport and dispersion model and/or READY website (https://www.ready.noaa.gov) used in this publication. We acknowledge the use of data and/or imagery from NASA's Fire Information for Resource Management System (FIRMS) (https://earthdata.nasa.gov/firms), part of NASA's Earth Observing System Data and Information System (EOSDIS). The data used are the standard fire products from the Moderate Resolution
Imaging Spectroradiometer (MODIS) from the Terra and Aqua platforms. Further, we acknowledge use of the MISR Enhanced Research





and Lookup Interface (MERLIN). MISR is the acronym for the multi-angle imaging spectroradiometer on the Terra satellite launched by NASA on 18 December 1999.

The article processing charges for this open-access publication were covered by the Karlsruhe Institute of Technology (KIT).



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





**Table 1.** MIPAS data versions used in the presented analysis. FR: full resolution measurement mode, RR: reduced resolution measurement mode, V5H and V5R: spectra version.

| Trace gas | FR, nominal mode | RR, nominal mode | RR, UTLS-1 mode |
|---|---|---|---|
| HCN | HCN_V5H_21 | HCN_V5R_222/223 | HCN_V5R_120 |
| CO | CO_V5H_20 | CO_V5H_220/221 | CO_V5R_120 |
| $C_2H_2$ | C2H2_V5H_20 | C2H2_V5R_221/222 | C2H2_V5R_120 |
| $C_2H_6$ | C2H6_V5H_20 | C2H6_V5R_220/221 | C2H6_V5R_120 |
| PAN | PAN_V5H_21 | PAN_V5R_222/223 | PAN_V5R_121 |
| HCOOH | HCOOH_V5H_20 | HCOOH_V5R_220/221 | HCOOH_V5R_120 |

**Table 2.** Vertical resolution (VR) in terms of full-width at half-maximum of the rows of the averaging kernels and estimated standard deviation (ESD) of the different pollutants at the altitudes of 8 km in the northern extra-tropics and of 14 km in the southern tropics.

| Trace gas | VR@8km / km | VR@14km / km | ESD@8km / pptv | ESD@14km / pptv |
|---|---|---|---|---|
| HCN | 4.5 | 4.5 | 28 | 22 |
| CO | 4.2 | 4.3 | $10^{(a)}$ | $22^a$ |
| $C_2H_2$ | 2.6 | 3.2 | 18 | 12 |
| $C_2H_6$ | 3.3 | 4.3 | 127 | 97 |
| PAN | 2.3 | 2.8 | 23 | 17 |
| HCOOH | 3.0 | 3.5 | 23 | 20 |

a value given in ppbv





**Table 3.** Upper section: Emission factors (g kg$^{-1}$) for species emitted from different types of biomass burning as given by Andreae (2019). The values in parentheses are the standard deviations between different measurements. Lower section: Corresponding emission ratios (pptv/pptv) with standard deviations, calculated by error propagation, in parentheses. The ratios were calculated under consideration of the molar mass ratios.

| Species | Savanna/ grassland | Tropical forest | Temperate forest | Boreal forest | Peat fires | Agric. residues | Biofuels w/o dung | Dung |
|---|---|---|---|---|---|---|---|---|
| CO | 69(20) | 104(39) | 113(50) | 121(47) | 260(23) | 76(55) | 83(29) | 89(42) |
| HCN | 0.44(0.26) | 0.44(0.21) | 0.64(0.39) | 0.53(0.30) | 4.4(1.2) | 0.42(0.18) | 0.39(0.17) | 1.27(0.73) |
| $C_2H_2$ | 0.31(0.29) | 0.35(0.39) | 0.31(0.09) | 0.28(0.13) | 0.11(0.05) | 0.27(0.24) | 0.68(0.37) | 0.68(0.41) |
| $C_2H_6$ | 0.42(0.32) | 0.88(0.23) | 0.69(0.56) | 0.97(0.37) | 1.85(0.35) | 0.79(0.62) | 0.63(0.61) | 1.28(0.70) |
| HCOOH | 0.21(0.13) | 0.49(0.28) | 0.91(1.18) | 1.04(0.89) | 0.29(0.14) | 0.56(0.45) | 0.23(0.22) | 0.39(0.05) |
| PAN | - | - | - | - | - | - | - | - |
| $\Delta$HCN/$\Delta$CO[a] | 0.66(0.44) | 0.44(0.27) | 0.59(0.44) | 0.45(0.31) | 1.75(0.50) | 0.57(0.48) | 0.49(0.27) | 1.48(1.10) |
| $\Delta C_2H_2$/$\Delta$HCN | 0.73(0.81) | 0.83(1.00) | 0.50(0.34) | 0.55(0.40) | 0.03(0.01) | 0.67(0.66) | 1.81(1.26) | 0.56(0.46) |
| $\Delta C_2H_6$/$\Delta$HCN | 0.86(0.83) | 1.80(0.98) | 0.97(0.98) | 1.65(1.12) | 0.38(0.13) | 1.69(1.51) | 1.45(1.54) | 0.91(0.72) |
| $\Delta$HCOOH/$\Delta$HCN | 0.28(0.24) | 0.65(0.49) | 0.83(1.20) | 1.15(1.18) | 0.04(0.02) | 0.78(0.71) | 0.35(0.36) | 0.18(0.11) |
| $\Delta$PAN/$\Delta$HCN | - | - | - | 0.9-2.75[b] | - | - | - | - |

a $\Delta$HCN/$\Delta$CO emission ratios are presented in %.
b Derived from values in Tereszchuk et al. (2013b).

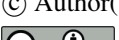



**Table 4.** Mean enhancement ratios (pptv/pptv) calculated from MIPAS measurements in different global regions (see Fig.8), presented for a minimum number of at least 4 bins. The thresholds for inclusion of individual $5° \times 15°$ bins for calculation of regional mean ERs are: 80 ppbv for CO, 60 pptv for $C_2H_2$, 300 pptv for $C_2H_6$, 100 pptv for PAN and 60 pptv for HCOOH. Further criteria: number of points per bin $\geq 10$ and correlation coefficient $r \geq 0.6$ ($r \geq 0.55$ for $\Delta HCN/\Delta CO$). The numbers in parentheses indicate the 1-$\sigma$ standard deviations of the bin averages. The $\lambda$-values used for Deming regression analysis are $4.42 \times 10^{-6}$ for $\Delta HCN/\Delta CO$, 0.35 for $\Delta C_2H_2\Delta/HCN$, 20 for $\Delta C_2H_6/\Delta HCN$, 0.63 for $\Delta PAN/\Delta HCN$ and 0.75 for $\Delta HCOOH/\Delta HCN$.

| Region | NAM | SIB | NHSUB | NAF | SAM | SAF | IOAUS | SMIDL |
|---|---|---|---|---|---|---|---|---|
| $\Delta HCN/\Delta CO^a$ | | | | | | | | |
| Feb | - | - | 0.25(0.06) | 0.18(0.03) | - | - | - | - |
| Apr | 0.25(0.08) | 0.33(0.15) | 0.29(0.06) | 0.15(0.03) | - | 0.18(0.05) | - | - |
| Jul | 0.29(0.11) | 0.28(0.11) | 0.20(0.07) | - | - | 0.22(0.02) | - | - |
| Aug | 0.39(0.13) | 0.42(0.21) | 0.23(0.08) | - | - | 0.30(0.06) | - | - |
| Oct | - | - | - | 0.38(0.22) | 0.43(0.13) | 0.50(0.09) | 0.65(0.13) | 0.68(0.18) |
| $\Delta C_2H_2\Delta/HCN$ | | | | | | | | |
| Feb | - | - | 1.83(0.37) | 0.80(0.17) | - | 0.95(0.19) | - | - |
| Apr | 1.60(0.46) | 1.59(0.32) | 1.00(0.31) | 0.61(0.10) | - | 0.94(0.31) | - | - |
| Jul | 0.73(0.22) | 0.86(0.32) | 1.14(0.28) | 1.03(0.21) | - | 0.98(0.22) | - | - |
| Aug | 0.64(0.23) | 0.57(0.30) | 1.18(0.33) | 1.06(0.23) | - | 0.77(0.17) | - | 0.58(0.17) |
| Sep | 0.53(0.15) | 0.60(0.39) | - | 1.11(0.44) | 0.42(0.11) | 0.56(0.18) | - | 0.47(0.11) |
| Oct | 1.50(0.51) | - | - | 0.43(0.18) | 0.33(0.05) | 0.32(0.04) | 0.20(0.09) | 0.33(0.07) |
| $\Delta C_2H_6/\Delta HCN$ | | | | | | | | |
| Feb | - | - | 5.87(2.85) | - | - | - | - | - |
| Apr | 6.09(3.33) | 6.67(2.23) | 2.94(1.96) | 1.72(0.28) | - | - | - | - |
| Jul | 3.81(1.27) | 3.18(1.02) | - | - | - | - | - | - |
| Aug | 2.37(0.72) | 2.37(0.83) | 2.49(1.67) | - | - | 1.04(0.19) | - | 0.97(0.09) |
| Sep | 3.05(1.06) | 2.49(0.68) | - | - | 1.08(0.22) | 1.08(0.52) | - | 1.08(0.20) |
| Oct | 4.74(1.28) | - | - | 1.14(0.34) | 0.93(0.11) | 0.86(0.09) | 0.64(0.22) | 0.95(0.13) |
| $\Delta PAN/\Delta HCN$ | | | | | | | | |
| Feb | - | - | 1.22(0.22) | 1.78(0.28) | 1.62(0.33) | 1.69(0.26) | - | 1.63(0.21) |
| Apr | 2.71(1.16) | - | 1.62(0.28) | 2.03(0.74) | 1.27(0.20) | 2.12(0.77) | 1.46(0.32) | 1.70(0.39) |
| Jul | 3.06(0.78) | 2.77(0.89) | 2.66(0.91) | 1.93(0.54) | 1.09(0.09) | 1.16(0.17) | - | - |
| Aug | 2.24(0.70) | 2.64(0.88) | 2.39(0.76) | 1.85(0.57) | 1.65(0.44) | 1.16(0.18) | - | 1.26(0.24) |
| Oct | 2.36(0.69) | 2.51(0.73) | 1.48(0.57) | 1.55(0.37) | 1.09(0.27) | 1.00(0.28) | 0.61(0.20) | 0.76(0.16) |
| $\Delta HCOOH/\Delta HCN$ | | | | | | | | |
| Feb | - | - | 1.09(0.35) | 0.74(0.26) | - | 0.66(0.17) | - | - |
| Apr | 0.84(0.23) | - | 0.77(0.23) | 0.57(0.15) | - | - | - | - |
| Jul | 1.61(0.46) | 1.55(0.50) | 1.16(0.99) | - | - | - | - | - |
| Aug | 1.04(0.60) | 1.23(0.46) | 0.89(0.29) | - | 1.21(0.26) | 0.80(0.64) | - | 1.85(0.74) |
| Sep | 0.95(0.51) | 1.17(0.80) | - | - | 0.85(0.32) | 0.57(0.23) | - | 1.16(0.43) |
| Oct | 0.84(0.23) | 1.13(0.51) | - | - | 0.52(0.20) | 0.38(0.09) | 0.41(0.32) | 0.72(0.27) |

a Enhancement ratios for $\Delta HCN/\Delta CO$ are given in %.



**Figure 1.** Global distribution of the thermal tropopause derived from MIPAS temperature profiles for the periods March to May (top left), June to August (top right), September to November (bottom left) and December to February (bottom right). The data are bin-averages (5° lat × 15° lon, 7.5° lat × 15° lon at the poles) of the time period 2002 to 2012.







**Figure 2.** Left: Global HCN distribution measured by MIPAS at 3 km below the tropopause during **(a)** March to May, **(c)** June to August, **(e)** September to November and **(g)** December to February. The distributions are averaged over the time period 2002 to 2012. Right: Same as left, but for CO measured during **(b)** March to May, **(d)** June to August, **(f)** September to November and **(h)** December to February.



**Figure 3.** Global carbon emissions by fires in g m$^{-2}$ during March to May (top left), June to August (top right), September to November (bottom left) and December to February (bottom right), added up over the measurement period of MIPAS (July 2002–March 2012). The original data are monthly carbon emissions of the Global Fire Emissions Database, version 3.1, (GFEDv3.1) at 0.25° latitude by 0.25° longitude spatial resolution.







**Figure 4.** Left: Global $C_2H_2$ distribution measured by MIPAS at 3 km below the tropopause during **(a)** March to May, **(c)** June to August, **(e)** September to November and **(g)** December to February. The distributions are averaged over the time period 2002 to 2012. Right: Same as left, but for $C_2H_6$ measured during **(b)** March to May, **(d)** June to August, **(f)** September to November and **(h)** December to February.





**Figure 5.** Left: Global PAN distribution measured by MIPAS at 3 km below the tropopause during **(a)** March to May, **(c)** June to August, **(e)** September to November and **(g)** December to February. The distributions are averaged over the time period 2002 to 2012. Right: Same as left, but for HCOOH measured during **(b)** March to May, **(d)** June to August, **(f)** September to November and **(h)** December to February. Note that the HCOOH-distributions of September to November and December to February are at 5 km below the tropopause.





**Figure 6.** Time series of monthly mean **(a)** HCN, **(b)** CO, **(c)** $C_2H_2$, **(d)** $C_2H_6$, **(e)** PAN and **(f)** HCOOH VMRs in $10°$-wide latitude bands at the altitude of 3 km below the tropopause, based on the complete MIPAS data set.







**Figure 7.** Monthly composites of Pearson correlation coefficients $r$ resulting from correlation analyses between MIPAS measurements in 5°-wide latitude bands, between 50°S and 50°N at 3 km and at higher latitudes at 2 km below the tropopause. **(a)** HCN versus CO, **(b)** $C_2H_2$ versus HCN, **(c)** $C_2H_6$ versus HCN, **(d)** PAN versus HCN, **(e)** HCOOH versus HCN. Values lower than 0.2 or higher than 0.9 are also labelled in dark blue or dark red, respectively.



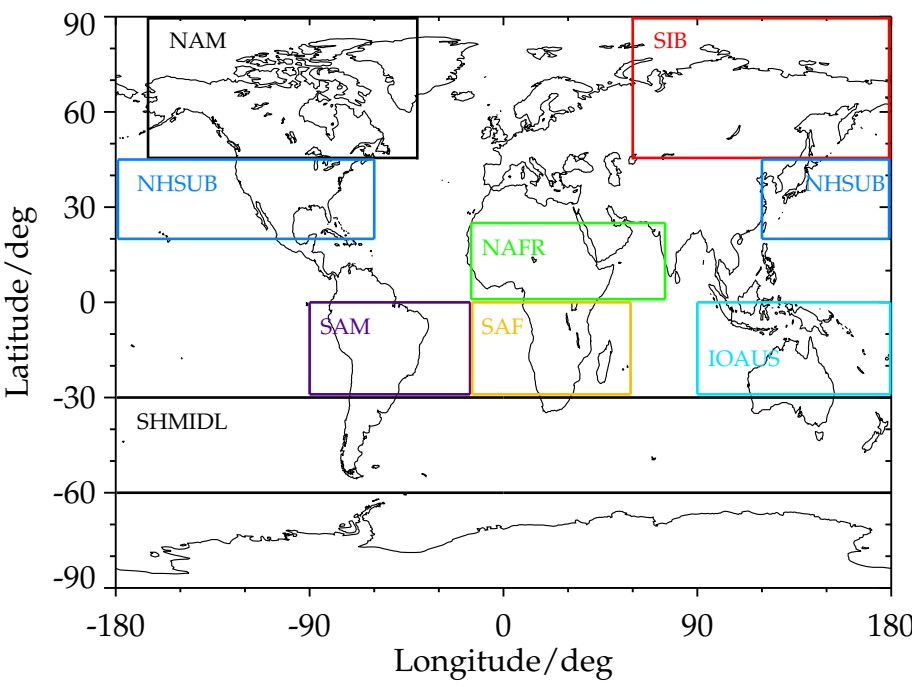

**Figure 8.** Global regions, for which the mean enhancement ratios shown in Tab. 4 and discussed in Sect. 4.3.2 were calculated. NAM: boreal North America, SIB: Siberia, NHSUB: Northern hemispheric subtropics, NAF: North Africa, SAM: South America, SAF: South Africa, IOAUS: Indonesia and Australia, SHMIDL: Southern hemispheric mid-latitudes.





**Figure 9.** Left: Global distribution of correlation coefficients *r* between MIPAS measurements of HCN and CO in the upper troposphere, for **(a)** February, **(c)** April, **(e)** July and **(g)** October. Right: Respective ΔHCN/ΔCO enhancement ratios (ER) in 0.01 pptv/pptv for **(b)** February, **(d)** April, **(f)** July and **(h)** October. The displayed altitude is 3 km below the tropopause in the latitude region 50°S to 50°N and 2 km below at higher latitudes. White areas are regions with correlation coefficients or enhancement ratios lower than zero, with CO VMRs lower than 80 ppbv or with bins containing less than 10 values. The bin-size is 5° latitude × 15° longitude. The distributions are monthly composites of the time period 2002 to 2012.




**Figure 10.** Left: Global distribution of correlation coefficients *r* between MIPAS measurements of $C_2H_2$ and HCN in the upper troposphere, for **(a)** February, **(c)** April, **(e)** August and **(g)** October. Right: Same as left, but for $\Delta C_2H_2/\Delta HCN$ enhancement ratios (ER) in pptv/pptv for **(b)** February, **(d)** April, **(f)** August and **(h)** October. White areas are regions with correlation coefficients or enhancement ratios lower than zero, with $C_2H_2$ VMRs lower than 60 pptv or with bins containing less than 10 values. For further details see Fig. 9.





**Figure 11.** Left: Global distribution of correlation coefficients *r* between MIPAS measurements of $C_2H_6$ and HCN in the upper troposphere, for **(a)** February, **(c)** April, **(e)** August and **(g)** October. Right: Same as left, but for $\Delta C_2H_6/\Delta HCN$ enhancement ratios (ER) in pptv/pptv for **(b)** February, **(d)** April, **(f)** August and **(h)** October. White areas are regions with correlation coefficients or enhancement ratios lower than zero, with $C_2H_6$ VMRs lower than 300 pptv or with bins containing less than 10 values. For further details see Fig. 9.



**Figure 12.** Left: Global distribution of correlation coefficients *r* between MIPAS measurements of PAN and HCN in the upper troposphere, for **(a)** February, **(c)** April, **(e)** July and **(g)** October. Right: Same as left, but for ΔPAN/ΔHCN enhancement ratios (ER) in pptv/pptv for **(b)** February, **(d)** April, **(f)** July and **(h)** October. White areas are regions with correlation coefficients or enhancement ratios lower than zero, with PAN VMRs lower than 100 pptv or with bins containing less than 10 values. For further details see Fig. 9.





**Figure 13.** Left: Global distribution of correlation coefficients *r* between MIPAS measurements of HCOOH and HCN in the upper tropo-sphere, for **(a)** February, **(c)** April, **(e)** August and **(g)** October. Right: Same as left, but for ΔHCOOH/ΔHCN enhancement ratios (ER) in pptv/pptv for **(b)** February, **(d)** April, **(f)** August and **(h)** October. White areas are regions with correlation coefficients lower than zero, with HCOOH VMRs lower than 60 pptv or with bins containing less than 10 values. For further details see Fig. 9.



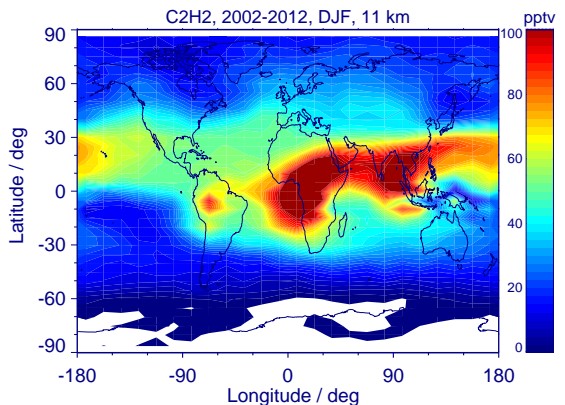 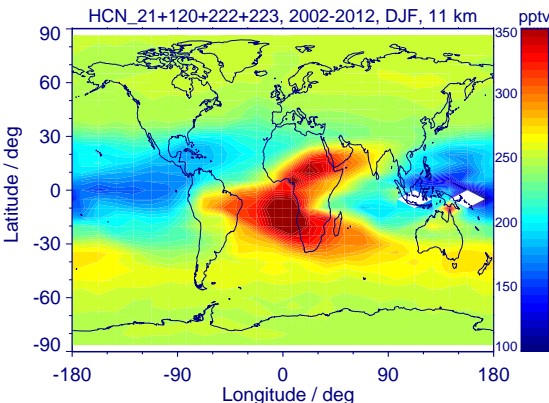

**Figure A1.** Global distributions of $C_2H_2$ (left) and of HCN (right) measured by MIPAS at 11 km altitude during December to February, averaged over the time period 2002 to 2012.





**Figure A2.** Global distributions of **(a)** $C_2H_6$, **(b)** $C_2H_2$, **(c)** HCN, **(d)** PAN and **(e)** HCOOH measured by MIPAS at 10 km altitude during July, averaged over the time period 2002 to 2012.



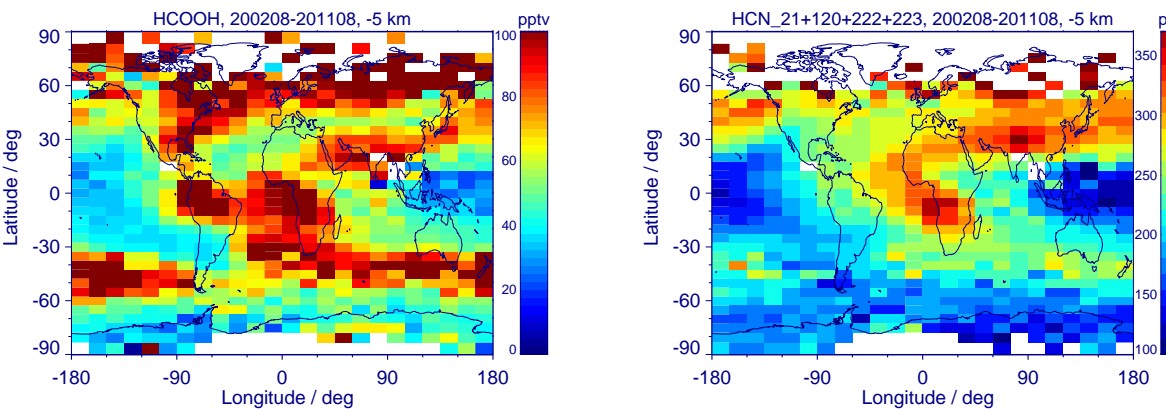

**Figure A3.** Global distributions of HCOOH (left) and of HCN (right) measured by MIPAS at 5 km below the tropopause during the August, averaged over the time period 2002 to 2012.