# Peer review of "Upper tropospheric pollutants observed by MIPAS: geographic and seasonal variations"

_EGUsphere, 2024_

## Referee Comment (RC1)

This article investigates average seasonal enhancements in pollutants related to biomass burning using measurements from the MIPAS instrument.  Ratios of the VMRs for different molecules are evaluated, relating relative enhancements of different constituents compared to enhancements in HCN to expectations for emission ratios from various types of vegetation, identifying when unexpectedly high levels relative to HCN suggest sources other than biomass burning.  In general, it is well written and thorough, a good summary of what MIPAS sees for this class of molecule.  However, there were a few issues that I felt should be addressed.

Firstly, I disagree with setting the tropopause height to 14.7 km for measurements within the polar vortex where the temperature minimum is around 20 km.  I agree that the assumptions associated with the WMO definition of tropopause have broken down in that situation, but a tropopause of 14.7 km seems way too high in this latitude region.  This can be illustrated by looking at the chemical makeup in the vicinity of 14.7 km within the Antarctic polar vortex.

[Figure]

Above: the temperature profile from occultation ss108064 from the Atmospheric Chemistry Experiment Fourier transform spectrometer (ACE-FTS), measured September 1, 2023, near latitude 80 °S.  Applying the WMO definition of tropopause would yield a value near 20 km.

[Figure]

Above: the $O_3$ volume mixing ratio (VMR) profile for ss108064. In the troposphere, $O_3$ levels should be relatively low. Judging from the above profile, the tropopause at the location of this occultation is somewhere around 10 km. 14.7 km is well into the stratosphere. Unfortunately, the altitude resolution of $O_3$ measurements from MIPAS presumably means $O_3$ cannot be used to estimate tropopause height.

The impact of this issue is probably best seen in the distribution plots for $C_2H_6$. For seasons where high average tropopause heights are seen in the data set, low levels of $C_2H_6$ are indicated in the distribution plots. This is not a chemical (i.e., low $C_2H_6$ in the troposphere) effect. Instead, it is a consequence of including stratospheric air (with low levels of $C_2H_6$) in the average.

[Figure]

[Figure]

The obvious solution would be to use a dynamical tropopause in the analysis rather than the WMO definition.  When dealing with measurements inside the polar vortex, it is possible that using a single boundary on potential vorticity (available from MERRA-2) might not suffice, and you would need to use a more sophisticated approach (doi:10.1029/2010JD014343, "Dynamical tropopause based on isentropic potential vorticity gradients" by Kunz et al., JGR Atmospheres, 2011).  The alternative would be to simply exclude Antarctic measurement where the WMO tropopause is above some threshold (15 km?).  Some distribution plots would likely end up with blank regions, but that would be preferable to reporting mostly stratospheric results as tropospheric.  Regions with no data would presumably all be in the Antarctic, which is the least interesting region anyway (scientifically speaking), with no significant sources of these molecules.

The second issue I had related to internal consistency for the different molecules.  Enhancement ratios are considered for each molecule separately, which admittedly already makes for a large amount of information to digest, but no synergy between the different molecules was considered.  For example, unexpectedly low enhancement ratios were observed for $C_2H_2$ and $C_2H_6$ in the tropics/subtropics in October.  The quite logical suggestion was made that this indicated aged emissions, with $C_2H_2$ and $C_2H_6$ levels having decreased more than HCN because of their shorter lifetimes.  However, the following observation was then made for HCOOH in October (page 20, line 26):

Thus both the high correlations as well as the ERs indicate that biomass burning is a major source of the enhanced HCOOH observed in the southern tropics and subtropics during this period.

The conclusion from $C_2H_2$ and $C_2H_6$ suggested that on average in October, the fires contributing to their enhancement in this region occurred a long time previously, long enough for the shorter lifetimes relative to HCN to have a significant effect.  Then the conclusion for HCOOH in October was the opposite, that there were "fresh" fires contributing to the HCOOH enhancement.  For me, it would have been more logical to conclude that the contributions to HCOOH enhancement from biomass burning

should have been relatively low because we generally have aged emissions in October (as suggested by the $C_2H_2$ and $C_2H_6$ observations), so the higher enhancement ratios for HCOOH (compared to those observed for $C_2H_2$ and $C_2H_6$) indicate biogenic sources are also likely contributing. This would be consistent with the conclusions in the paper suggesting biogenic sources for HCOOH during other time periods. Why would there be biogenic sources of HCOOH during other time periods but not this one?

As the authors indicate, the different vertical resolutions HCN and the other molecules will impact the accuracy of the ratio. They report that degrading $C_2H_2$ to HCN resolution yields "on average" 10% lower $C_2H_2$. It was not clear what conditions were being averaged over. Over the global set? The vertical resolution changes with latitude, and so does the tropopause height, which could compound the systematic error. Since the vertical resolution for $C_2H_2$ is larger at 14 km than it is at 8 km, presumably the impact would be smaller in the tropics than near the poles because there would be less of a resolution mismatch.

Regardless, an error of 10% is probably low compared to the uncertainty from the unknown "emission age" of the air, due to the different lifetimes of the molecules being ratioed. That could perhaps be stressed more when comparing observations of enhancement ratios to emission ratios for various types of vegetation. It is difficult to make definitive statements of agreement/disagreement with the expectations for a particular type of vegetation without knowing how long ago the fire occurred.

Minor comments:

Page 7, line 6: but ca also be caused

ca -> can

Page 7, line 16: caused by confinement of pollution mostly from South- and South-East Asia (see, e.g., Vogel et al., 2015) inside the Asian Monsoon Anticyclone (AMA)

Not just confinement; vertical transport within the monsoon region carries pollution toward the upper troposphere, increasing the measured enhancement for a point 2 km below the tropopause.

Sometimes "Table" is written out fully and sometimes it is abbreviated as "Tab."

Figure 8: North Africa is labelled as NAFR but is referred to as NAF in the text

Page 17, line 26: Like for HCH versus CO

HCH -> HCN

Page 19: no reference to Figure 13 in the text

---

## Author Comment (AC2)

**Response to the comments of reviewer #2 on the paper "Upper tropospheric pollutants observed by MIPAS: geographic and seasonal variations", egusphere-2024-1793**

Norbert Glatthor et al.

Reviewer comments are in black, while our replies are in blue.

Overall, this is an interesting study making use of long-term MIPAS retrievals to investigate the potential sources controlling the spatio-temporal distribution of upper tropospheric pollutants. This study exploits information from enhancement ratios (ER) in comparison to emission ratios from known source types (e.g. forest fires) to determine the likely source of the pollutants. The manuscript is generally well written, figure presentation is good and would be an interesting addition to ACP. Therefore, subject to some minor comments, it is suitable for publication in ACP.

We thank reviewer 2 for this positive assessment.

1. The GFED data used in Figure 3 is from an older version (3.1). Therefore, it makes sense to exploit a newer version e.g. (vn4.1s or vn5).

We will exploit GFED version 5. The respective plots have already been created, and there are only slight differences to the original Figure 3 of the manuscript. Thus, except of referring to version 5 instead of version 3.1, the wording in the manuscript has not to be adjusted.

2. The authors use the ERs to infer source information about the pollutants retrieved by MIPAS. However, in places these statements are overall "conclusive" and need to be weakened (e.g. Page 17 Line 33, Page 20 Line 7 and Page 9 Line 20, Page 12 Lines 21–22) without the support of e.g. a model.

On page 17, lines 29–32 we exclude biomass burning as major source of C2H6 from inspection of the observed correlation coefficients and ERs. Then we conclude on lines 32-33 that "Both findings point to other sources of enhanced C2H6 at northern mid- and high latitudes during these months, namely accumulation in boreal winter due to the minimum in the OH cycle and anthropogenic activities." We think that these conclusions are rather cautious and list the only remaining processes for C2H6 production during these times of the year. Thus we do not see the need to weaken this statement.

The same applies to the text passage on page 20, line 7. Here we write "There are only low to moderate correlations north of 40N, indicating that the northern hemisperic HCOOH increase in spring (cf. Fig. 5) is not caused by biomass burning but rather by oxidation of biogenic precursors."

Page 9, line 20: We will change the sentence "In austral winter (JJA) the amounts increase slightly at southern mid-latitudes, which obviously is also due to the OH cycle." into "In austral winter (JJA) the amounts increase slightly at southern mid-latitudes, which might also be caused by the minimum in the OH cycle."

Page 12, lines 20–22: "From October to March the C2H2 and HCN amounts at northern high latitudes are uncorrelated, confirming the dominant influence of the OH cycle and of biofuel burning on the C2H2 amounts in these regions during winter." We draw these conclusions because we think that the dominant influence of the OH cycle and of biofuel burning on the C2H2 amounts in these regions during winter is well approved in the literature. For illustration we already give a reference to Zander et al. (1991) on page 8, line 21. We suggest to add further references (e.g., Goldstein et al., 1995; Xiao et al., 2007) at this passage of the manuscript.

Goldstein, A.H., et al.:, "Seasonal variations of nonmethane hydrocarbons in rural New England: Constraints on OH concentrations in northern midlatitudes", J. Geophys. Res., https://doi.org/10.1029/95JD02034, 1995.

Xiao et al, 2007: see reference list.

3. The methods used in Section 4.3.2 need some more detail. From reading the text (Page 13 Lines 15–22), it is not overly clear what ESDs are and how they are calculated and why $\lambda$ is needed in the regression model.

The term ESD is introduced as estimated standard deviation on page 4, line 30. We will further outline at this text passage, that the ESD is "measurement noise transformed into parameter space by the retrieval program". The $\lambda$-term is used as a weighting parameter in the regression model to take the generally different uncertainties of the two pollutants into account. To give some more details, we will change the text passage beginning at page 13, line 18:

"In case of equal error variances for both variables, Deming regression simplifies to orthogonal regression. The $\lambda$-values required for Deming regression were determined from the ESDs given in Tab. 2 as = $ESD_y^2/ESD_x^2$. After averaging over the two height levels in Tab. 2, the following global $\lambda$-values were applied: $\Delta HCN/\Delta CO$: $4.42\times10^{-6}$, $\Delta C_2H_2/\Delta HCN$: 0.35, $\Delta C_2H_6/\Delta HCN$: 20.01, $\Delta PAN/\Delta HCN$: 0.63 and $\Delta HCOOH/\Delta HCN$: 0.75.

into

"In this regression model the potentially different uncertainties $\sigma_1$ and $\sigma_2$ of both pollutants are taken into account by introducing a parameter $\lambda = \sigma_1^{-2} \times \sigma_2^2$. We determined the $\lambda$-values from the ESDs given in Table 2 as $\lambda = ESD_1^{-2} \times ESD_2^2$. After averaging over the two height levels in Table 2, the following global $\lambda$-values were applied: $\Delta HCN/\Delta CO$: $4.42\times10^{-6}$, $\Delta C_2H_2/\Delta HCN$: 0.35, $\Delta C_2H_6/\Delta HCN$: 20.01, $\Delta PAN/\Delta HCN$: 0.63, and $\Delta HCOOH/\Delta HCN$: 0.75. In case of equal error variances for both variables, Deming regression simplifies to orthogonal regression.".

4. In Section 4.3.2, why focus on individual months instead of the seasons as used earlier in the manuscript and why choice those specific months (i.e. Feb, Apr, Jul, Oct)?

In Section 4.3.2 we focus on individual months, because the differences between late boreal winter (Feb) and boreal spring (Apr) in the northern hemisphere as well as the differences between austral winter (Jul/Aug) and austral spring (Oct) in the

southern hemisphere become more obvious in this representation. For these reasons we do not want to change the discussion to seasons here.

Page 9 Line 17: Reword "even better visible" to something like "more evident".

5 We will change "even better visible" to "more evident".

Page 9 Line 18: Can you provide a motivation for looking at 11 km (e.g. why not 10 km)?

Actually the global distributions of C2H6 and of the other gases in Figure A2 are presented at 10 km altitude. "11 km" is a typo, which will be corrected. For the reason of consistency, the C2H2 and HCN enhancements in Figure A1, which are

10 presented at 11 km, will also be presented at 10 km.

Page 16 Lines 14-19: Suggest rewording as not overly easy to follow.

We will change the wording from

"High correlations between C2H2 and HCN occur in even larger regions than for HCN and CO and they are generally stronger,

15 often reaching values of $r = 0.8$ or above (Fig. 10, left column). Again, the focus is above North-East Africa, the southern tropical Atlantic and the North Pacific in February and April. In August, there are high correlations at northern mid- and high latitudes, above tropical Africa, in the AMA region and above the southern Indian Ocean, and in October very high correlations above the southern tropical Atlantic, South Africa, the Indian Ocean and Australia. All these features strongly point to biomass burning as common source of C2H2 and HCN."

20 into

"High correlations between C2H2 and HCN occur in even larger regions than those between HCN and CO and they are generally stronger, often reaching values of $r = 0.8$ or above (Fig. 10, left column). Again, strong correlations are found above North-East Africa, the southern tropical Atlantic and the North Pacific in February and April, and at northern mid- and high latitudes, above tropical Africa, in the AMA region and above the southern Indian Ocean in August. In October there are

25 very high correlations above the southern tropical Atlantic, South Africa, the Indian Ocean and Australia. All these features strongly point to biomass burning as common source of C2H2 and HCN."

Page 16 Line 28: Add "a" between "as" and "possible".

Will be done.

30

Page 17 Line 28: This line needs rewording as not clearly written. Also, what is the "two times larger relative retrieval error" based on? Is this from the literature, the data product, do you calculate this?

This value was calculated from the ESDs associated to the retrieved VMRs. For more clarity we will change the sentence

"But it has to be considered that a certain part of the weaker correlations is due to the up to two times larger relative retrieval

35 error of C2H6 as compared to C2H2."

into

"But it has to be considered that a certain part of the weaker correlations is due to the large retrieval error of C2H6 (see Table 2), which results in an up to two times larger relative retrieval error of C2H6 (error divided by VMR) as compared to C2H2."

5   Page 18 Line 33: Add "any" between "hardly" and "information".

Will be done.

Page 21 Line 8–9: This sentence needs rewording as difficult to follow.

The sentence

10   "The reason is the minimum of their dominant reactant of C2H2 and C2H6, the OH radical, as well as enhanced biofuel and fossil fuel burning during winter."

will be changed into

"The reason is the minimum of their dominant reactant, the OH radical, as well as enhanced biofuel and fossil fuel burning during winter."

15

Page 21 Line 22: Add "an" between "as" and "important".

Will be done.

Page 21 Line 26: Replace "better" with e.g. "stronger".

20   Will be done.

Page 22 Line 24: Replace "become better" with "becomes more".

Will be done.

25   Table 2: For "a" why give the value as ppbv. For consistency and presentation, keep as pptv.

Ok, we will give the values for CO in pptv.

Table 3: Same as Table 2, why use "a"? Best to use consistent units and not a %.

Ok, we will specify the HCN/CO enhancement ratios in Table 3 consistently in pptv/pptv. We will perform the same update in

30   Table 4 as well.

Figure 6: Panel title and x/y-axes are missing for PAN.

We do not see any differences between the depiction of PAN and of the other gases in Figure 6.

35   Figure 8: I do not understand the delta HCN / delta CO units of 0.01 pptv/pptv while the colour bars show ER %.

The reviewer obviously means Figure 9? Consistently to the changes requested above, we will plot the HCN/CO enhancement ratios in Figure 9 in pptv/pptv as well. Thus, "0.01 pptv/pptv" will be changed into "pptv/pptv" in the figure captions, and the colour bar units will also be changed from "%" into "pptv/pptv."

Figure 3: For the colour bar units, what is 10 to the power of? Just says "10".

5  In the original Figure 3 the numbers at the colour bar should have been added as exponents to "10". To make things clearer, we will update Figure 3, with the numbers $10^{-2}$, $10^{-1}$, ..., $10^{3}$ at the left side and the unit g m$^{-2}$ at the top of the colour bar.

---

## Author Response (AR1)

In the following, please find the reviewer comments in black and our replies and performed changes in blue.

**Response to the comments of reviewer #1:**

This article investigates average seasonal enhancements in pollutants related to biomass burning using measurements from the MIPAS instrument. Ratios of the VMRs for different molecules are evaluated, relating relative enhancements of different constituents compared to enhancements in HCN to expectations for emission ratios from various types of vegetation, identifying when unexpectedly high levels relative to HCN suggest sources other than biomass burning. In general, it is well written and thorough, a good summary of what MIPAS sees for this class of molecule. However, there were a few issues that I felt should be addressed.

We thank Chris Boone for this overall positive assessment.

Firstly, I disagree with setting the tropopause height to 14.7 km for measurements within the polar vortex where the temperature minimum is around 20 km. I agree that the assumptions associated with the WMO definition of tropopause have broken down in that situation, but a tropopause of 14.7 km seems way too high in this latitude region. This can be illustrated by looking at the chemical make up in the vicinity of 14.7 km within the Antarctic polar vortex.
The impact of this issue is probably best seen in the distribution plots for C2H6. For seasons where high average tropopause heights are seen in the data set, low levels of C2H6 are indicated in the distribution plots. This is not a chemical (i.e., low C2H6 in the troposphere) effect. Instead, it is a consequence of including stratospheric air (with low levels of C2H6) in the average. The obvious solution would be to use a dynamical tropopause in the analysis rather than the WMO definition. When dealing with measurements inside the polar vortex, it is possible that using a single boundary on potential vorticity (available from MERRA-2) might not suffice, and you would need to use a more sophisticated approach (doi:10.1029/2010JD014343, "Dynamical tropopause based on isentropic potential vorticity gradients" by Kunz et al., JGR Atmospheres, 2011). The alternative would be to simply exclude Antarctic measurement where the WMO tropopause is above some threshold (15 km?). Some distribution plots would likely end up with blank regions, but that would be preferable to reporting mostly stratospheric results as tropospheric. Regions with no data would presumably all be in the Antarctic, which is the least interesting region anyway (scientifically speaking), with no significant sources of these molecules.

The reviewer is right. Setting the tropopause height to 14.7 km for geolocations in the antarctic vortex, where the WMO definition of tropopause has broken down, was no good solution. We followed his second suggestion and excluded measurements at latitudes south of 50°S, for which during June to November a tropopause height of more than 13.5 km was determined (cf. Fig. S1 in the Supplement). We applied the same height criterion for the region north of 50°N during December to February, since some unplausible high tropopause heights were also determined for measurements in the arctic vortex (cf. Fig. S1). The differences of the updated global distributions of the pollutants to the previous plots are generally small and only visible for antarctic C2H6 during June–August and September–November. Especially the anticorrelation of the C2H6 mixing ratios to tropopause height during Antarctic spring (Fig. 4f) has largely disappeared. Like in the original plots there are no blank regions (no data) in the global distributions for spring, autumn and winter. The small blank regions in the Antarctic during June-August are only marginally enlarged.

The second issue I had related to internal consistency for the different molecules. Enhancement ratios are considered for each molecule separately, which admittedly already makes for a large amount of information to digest, but no synergy between the different molecules was considered. For example, unexpectedly low enhancement ratios were observed for C2H2 and C2H6 in the tropics/subtropics in October. The quite logical suggestion was made that this indicated aged emissions, with C2H2 and C2H6 levels having decreased more than HCN because of their shorter lifetimes. However, the following observation was then made for HCOOH in October (page 20, line 26):
"Thus both the high correlations as well as the ERs indicate that biomass burning is a major source of the enhanced HCOOH observed in the southern tropics and subtropics during this period."

The conclusion from C2H2 and C2H6 suggested that on average in October, the fires contributing to their enhancement in this region occurred a long time previously, long enough for the shorter lifetimes relative to HCN to have a significant effect. Then the conclusion for HCOOH in October was the opposite, that there were "fresh" fires contributing to the HCOOH enhancement. For me, it would have been more logical to conclude that the contributions to HCOOH enhancement from biomass burning should have been relatively low because we generally have aged emissions in October (as suggested by the C2H2 and C2H6 observations), so the higher enhancement ratios for HCOOH (compared to those observed for C2H2 and C2H6) indicate biogenic sources are also likely contributing. This would be consistent with the conclusions in the paper suggesting biogenic sources for HCOOH during other time periods. Why would there be biogenic sources of HCOOH during other time periods but not this one?

In principle we definitely wanted to consider the synergy between the different molecules in the discussion in Section 4.3.2. For this reason we added some more cross references related to the enhancement ratios obtained for the different pairs of pollutants. However, we disagree with some of the other statements of the reviewer performed above. We do not report on unexpectedly low tropical/subtropical C2H6/HCN enhancement ratios in October but state that these ERs agree well with emission ratios for savanna and grassland burning (page 18, lines 20ff). Further, we do not intend to conclude from the low C2H2/HCN enhancement ratios of October that "the fires contributing to their enhancement in this region occurred a long time previously". Fig. 3 in our manuscript shows that the fire activity in southern tropical and subtropical Africa persists well into the period September–November and that the fire emissions in South America even peak during this period. Taking this into account we explain the low C2H2/HCN enhancement ratios in October with the sentence "Presumably this is an effect of plume aging, in this case stronger accumulation of HCN due to its relatively long lifetime (page 17, lines 15f)." To make our conclusion clearer, we changed this sentence into "Presumably this is an effect of plume aging, in this case stronger accumulation of HCN during persistent fire activity due to its relatively long lifetime."

Since the atmospheric lifetime of HCOOH is even somewhat shorter than that of C2H2, the reviewer rightly points out the discrepancy between our conclusions from the October ERs of C2H2/HCN ("Presumably this is an effect of plume aging ...") on page 17 lines 15f and our conclusions on HCOOH/HCN ("Thus both the high correlations as well as the ERs indicate that biomass burning is a major source of the enhanced HCOOH observed in the southern tropics and subtropics during this period.") on page 20, lines 26f. Therefore we changed this sentence into "Thus the high correlations indicate that biomass burning is a major source of the enhanced HCOOH observed in the southern tropics and subtropics during this period." and added the sentences "However, contrary to the other short-lived pollutant C2H2, the October ERs of HCOOH have not yet decreased considerably with respect to the emission ratios for savanna or tropical forest fires. This is an indication for additional contribution of biogenic release or of secondary production to the observed HCOOH amounts, giving the false impression of exclusively fresh pollution by biomass burning only."

As the authors indicate, the different vertical resolutions HCN and the other molecules will impact the accuracy of the ratio. They report that degrading C2H2 to HCN resolution yields "on average" 10% lower C2H2. It was not clear what conditions were being averaged over. Over the global set? The vertical resolution changes with latitude, and so does the tropopause height, which could compound the systematic error. Since the vertical resolution for C2H2 is larger at 14 km than it is at 8 km, presumably the impact would be smaller in the tropics than near the poles because there would be less of a resolution mismatch.

The value of 10% lower C2H2 was estimated from a restricted dataset of 154 MIPAS-orbits of January, February and October 2009, averaged over all latitudes. On behalf of the reviewer's comments, we performed additional estimations for different regions, e.g., the area 45°–90°N or the southern tropics and subtropics. The outcome was similar to our original estimation. We now give a more detailed estimation in the updated manuscript by changing the wording on page 14, lines 5-6, from "However, degradation of, e.g., the $C_2H_2$ profiles to the height resolution of HCN on average resulted in 10% lower $C_2H_2$ VMRs, which would lead to 10% lower ERs only."
into
"However, degradation of a sample of $C_2H_2$ profiles of 154 orbits to the height resolution of HCN by the use of Gaussians representing the differences in vertical resolution on average resulted in about 5% lower VMRs in the southern tropics and

subtropics and 10% lower VMRs at northern mid- and high latitudes, which would lead to 5–10% lower ERs only."

Regardless, an error of 10% is probably low compared to the uncertainty from the unknown "emission age" of the air, due to the different lifetimes of the molecules being ratioed. That could perhaps be stressed more when comparing observations of
enhancement ratios to emission ratios for various types of vegetation. It is difficult to make definitive statements of agreement / disagreement with the expectations for a particular type of vegetation without knowing how long ago the fire occurred.

We already mention the precautions to be taken with respect to the unknown "emission age" of the air on P13, L33ff. As suggested by the reviewer, we additionally point out this fact in the discussion of the ERs of the short-lived pollutants $C_2H_2$ and HCOOH with HCN by changing the passage on page 16, lines 20ff from
"The mean $\Delta C_2H_2/\Delta HCN$ enhancement ratio obtained for February for the region above northern tropical Africa and the Middle East is 0.80±0.17 (Fig. 10b). This value - also after a 10%-subtraction as discussed above - is in good agreement with the emission ratio for the dominant burning process in northern tropical Africa, namely savanna or grassland fires (0.73)."
into
"As already mentioned, the considerably shorter lifetime of $C_2H_2$ compared to HCN is an issue in the interpretation of monthly composites of $\Delta C_2H_2/\Delta HCN$ enhancement ratios. Nevertheless, the mean ER obtained for February for the region above northern tropical Africa and the Middle East is 0.80±0.17 (Fig. 10b), which - also after a 10%-subtraction as discussed above - is in good agreement with the emission ratio for the dominant burning process in northern tropical Africa, namely savanna or grassland fires (0.73)."
and by adding the sentence
"However, when discussing $\Delta HCOOH/\Delta HCN$ enhancement ratios, the same caveats have to be taken into accout as for $\Delta C_2H_2/\Delta HCN$."
on page 19, line 30 after "... enhanced HCOOH."

Minor comments:

Page 7, line 6: but ca also be caused, ca -> can

Has been corrected.

Page 7, line 16: "caused by confinement of pollution mostly from South- and South-East Asia (see, e.g., Vogel et al., 2015) inside the Asian Monsoon Anticyclone (AMA)."
Not just confinement; vertical transport within the monsoon region carries pollution toward the upper troposphere, increasing the measured enhancement for a point 2 km below the tropopause.

We changed the sentence into "This feature is caused by vertical transport of pollution mostly from South- and South-East Asia within the Asian monsoon region towards the upper troposphere and subsequent confinement inside the Asian Monsoon Anticyclone (AMA) (see, e.g., Vogel et al., 2015).

Sometimes "Table" is written out fully and sometimes it is abbreviated as "Tab."

We now write "Table" throughout the text.

Figure 8: North Africa is labelled as NAFR but is referred to as NAF in the text.

We now label North Africa as NAF in Fig. 8.

Page 17, line 26: Like for HCH versus CO, HCH -> HCN

has been corrected.

Page 19: no reference to Figure 13 in the text

5   We now begin the discussion "HCOOH versus HCN" with the sentence: "In Fig. 13 we show correlation coefcients and enhancement ratios for HCOOH versus HCN."

**Response to the comments of reviewer #2:**

10

Overall, this is an interesting study making use of long-term MIPAS retrievals to investigate the potential sources controlling the spatio-temporal distribution of upper tropospheric pollutants. This study exploits information from enhancement ratios (ER) in comparison to emission ratios from known source types (e.g. forest fires) to determine the likely source of the pollutants. The manuscript is generally well written, figure presentation is good and would be an interesting addition to ACP.
15  Therefore, subject to some minor comments, it is suitable for publication in ACP.

We thank reviewer 2 for this positive assessment.

1. The GFED data used in Figure 3 is from an older version (3.1). Therefore, it makes sense to exploit a newer version e.g.
20  (vn4.1s or vn5).
2. The authors use the ERs to infer source information about the pollutants retrieved by MIPAS. However, in places these statements are overall "conclusive" and need to be weakened (e.g. Page 17 Line 33, Page 20 Line 7 and Page 9 Line 20, Page 12 Lines 21–22) without the support of e.g. a model.

25  On page 17, lines 29–32 we exclude biomass burning as major source of C2H6 from inspection of the observed cor-relation coefficients and ERs. Then we conclude on lines 32-33 that "Both findings point to other sources of enhanced C2H6 at northern mid- and high latitudes during these months, namely accumulation in boreal winter due to the mini-mum in the OH cycle and anthropogenic activities." We think that these conclusions are rather cautious and list the only remaining processes for C2H6 production during these times of the year. Thus we do not see the need to weaken this statement.
30

The same applies to the text passage on page 20, line 7. Here we write "There are only low to moderate correlations north of 40N, indicating that the northern hemisperic HCOOH increase in spring (cf. Fig. 5) is not caused by biomass burning but rather by oxidation of biogenic precursors."
Page 9, line 20: We changed the sentence "In austral winter (JJA) the amounts increase slightly at southern mid-latitudes, which
35  obviously is also due to the OH cycle." into "In austral winter (JJA) the amounts increase slightly at southern mid-latitudes, which might also be caused by the minimum in the OH cycle."
Page 12, lines 20–22: "From October to March the C2H2 and HCN amounts at northern high latitudes are uncorrelated, confirming the dominant influence of the OH cycle and of biofuel burning on the C2H2 amounts in these regions during winter." We draw these conclusions because we think that the dominant influence of the OH cycle and of biofuel burning
40  on the C2H2 amounts in these regions during winter is well approved in the literature. For confirmation we already give a reference to Zander et al. (1991) on page 8, line 21. For further evidence we added further references to Goldstein et al. (1995) and to Xiao et al. (2007) at this passage of the manuscript.

Goldstein, A.H., et al.:, "Seasonal variations of nonmethane hydrocarbons in rural New England: Constraints on OH concen-
45  trations in northern midlatitudes", J. Geophys. Res., https://doi.org/10.1029/95JD02034, 1995.
Xiao et al, 2007: see reference list.

3. The methods used in Section 4.3.2 need some more detail. From reading the text (Page 13 Lines 15–22), it is not overly clear what ESDs are and how they are calculated and why $\lambda$ is needed in the regression model.

The term ESD is introduced as estimated standard deviation on page 4, line 30. We further outlined at this text passage, that the ESD is "measurement noise transformed into parameter space by the retrieval program". The $\lambda$-term is used as a weighting parameter in the regression model to take the generally different uncertainties of the two pollutants into account. To give some more details, we changed the text passage beginning at page 13, line 18:
"In case of equal error variances for both variables, Deming regression simplifies to orthogonal regression. The $\lambda$-values required for Deming regression were determined from the ESDs given in Tab. 2 as $= ESD_y^2/ESD_x^2$. After averaging over the two height levels in Tab. 2, the following global $\lambda$-values were applied: $\Delta HCN/\Delta CO$: $4.42\times10^{-6}$, $\Delta C_2H_2/\Delta HCN$: 0.35, $\Delta C_2H_6/\Delta HCN$: 20.01, $\Delta PAN/\Delta HCN$: 0.63 and $\Delta HCOOH/\Delta HCN$: 0.75.
into
"In this regression model the potentially different uncertainties $\sigma_1$ and $\sigma_2$ of both pollutants are taken into account by introducing a parameter $\lambda = \sigma_1^{-2} \times \sigma_2^2$. We determined the $\lambda$-values from the ESDs given in Table 2 as $\lambda = ESD_1^{-2} \times ESD_2^2$. After averaging over the two height levels in Table 2, the following global $\lambda$-values were applied: $\Delta HCN/\Delta CO$: $4.42\times10^{-6}$, $\Delta C_2H_2/\Delta HCN$: 0.35, $\Delta C_2H_6/\Delta HCN$: 20.01, $\Delta PAN/\Delta HCN$: 0.63, and $\Delta HCOOH/\Delta HCN$: 0.75. In case of equal error variances for both variables, Deming regression simplifies to orthogonal regression.".

4. In Section 4.3.2, why focus on individual months instead of the seasons as used earlier in the manuscript and why choice those specific months (i.e. Feb, Apr, Jul, Oct)?
In Section 4.3.2 we focus on individual months, because the differences between late boreal winter (Feb) and boreal spring (Apr) in the northern hemisphere as well as the differences between austral winter (Jul/Aug) and austral spring (Oct) in the southern hemisphere become more obvious in this representation. For these reasons we do not want to change the discussion to seasons here.

Page 9 Line 17: Reword "even better visible" to something like "more evident".

We changed "even better visible" to "more evident".

Page 9 Line 18: Can you provide a motivation for looking at 11 km (e.g. why not 10 km)?

Actually the global distributions of C2H6 and of the other gases in Fig. A2 are presented at 10 km altitude. "11 km" is a typo, which has been corrected. For the reason of consistency, the C2H2 and HCN enhancements in Fig. A1, which were presented at 11 km, are now also presented at 10 km.

Page 16 Lines 14-19: Suggest rewording as not overly easy to follow.

We changed the wording from
"High correlations between C2H2 and HCN occur in even larger regions than for HCN and CO and they are generally stronger, often reaching values of $r = 0.8$ or above (Fig. 10, left column). Again, the focus is above North-East Africa, the southern tropical Atlantic and the North Pacific in February and April. In August, there are high correlations at northern mid- and high latitudes, above tropical Africa, in the AMA region and above the southern Indian Ocean, and in October very high correlations above the southern tropical Atlantic, South Africa, the Indian Ocean and Australia. All these features strongly point to biomass burning as common source of C2H2 and HCN."
into
"High correlations between C2H2 and HCN occur in even larger regions than those between HCN and CO and they are generally stronger, often reaching values of $r = 0.8$ or above (Fig. 10, left column). Again, strong correlations are found above North-East Africa, the southern tropical Atlantic and the North Pacific in February and April, and at northern mid- and high latitudes, above tropical Africa, in the AMA region and above the southern Indian Ocean in August. In October there are

very high correlations above the southern tropical Atlantic, South Africa, the Indian Ocean and Australia. All these features strongly point to biomass burning as common source of C2H2 and HCN."

Page 16 Line 28: Add "a" between "as" and "possible".

Has been added.

Page 17 Line 28: This line needs rewording as not clearly written. Also, what is the "two times larger relative retrieval error" based on? Is this from the literature, the data product, do you calculate this?

This value was calculated from the ESDs associated to the retrieved VMRs. For more clarity we changed the sentence
"But it has to be considered that a certain part of the weaker correlations is due to the up to two times larger relative retrieval error of C2H6 as compared to C2H2."
into
"But it has to be considered that a certain part of the weaker correlations is due to the large retrieval error of C2H6 (see Table 2), which results in an up to two times larger relative retrieval error of C2H6 (error divided by VMR) as compared to C2H2."

Page 18 Line 33: Add "any" between "hardly" and "information".

Has been added.

Page 21 Line 8–9: This sentence needs rewording as difficult to follow.

The sentence
"The reason is the minimum of their dominant reactant of C2H2 and C2H6, the OH radical, as well as enhanced biofuel and fossil fuel burning during winter."
has been changed into
"The reason is the minimum of their dominant reactant, the OH radical, as well as enhanced biofuel and fossil fuel burning during winter."

Page 21 Line 22: Add "an" between "as" and "important".

Will be done.

Page 21 Line 26: Replace "better" with e.g. "stronger".

Has been done.

Page 22 Line 24: Replace "become better" with "becomes more".

Has been done.

Table 2: For "a" why give the value as ppbv. For consistency and presentation, keep as pptv.

The values for CO in Table 2 are now given in pptv.

Table 3: Same as Table 2, why use "a"? Best to use consistent units and not a %.

The HCN/CO enhancement ratios in Table 3 are now also given in pptv/pptv. For the reason of consistency, the same update for the HCN/CO ERs has been performed in Table 4 and in the discussion of "HCN versus CO" in Section 4.3.2 as well.

Figure 6: Panel title and x/y-axes are missing for PAN.

We do not see any differences between the depiction of PAN and of the other gases in Fig. 6.

Figure 8: I do not understand the delta HCN / delta CO units of 0.01 pptv/pptv while the colour bars show ER %.

The reviewer obviously means Fig. 9. Consistently to the changes requested above, we now plot the HCN/CO enhancement ratios in Fig. 9 in pptv/pptv as well. Thus, "0.01 pptv/pptv" has been changed into "pptv/pptv" in the figure captions, and the colour bar units have been changed from "%" into "pptv/pptv."

Figure 3: For the colour bar units, what is 10 to the power of? Just says "10".

In the original Fig. 3 the numbers at the colour bar should have been added as exponents to "10". To make things clearer, we updated Fig. 3, with the numbers $10^{-2}$, $10^{-1}$, ..., $10^3$ at the left side and the unit g m$^{-2}$ at the top of the colour bar.

**Additional changes:**

After removal of geolocations with unplausible antarctic and arctic tropopause heights in the data sets for the updated manuscript, changes in tropopause related mixing ratios and enhancement ratios are to be expected during winter and spring at high latitudes only. However, the updated enhancement ratios for $\Delta$HCN/$\Delta$CO, $\Delta$C$_2$H$_2$/$\Delta$HCN and $\Delta$HCOOH/$\Delta$HCN in Section 4.3.2 and in Table 4 also show slight differences for mid-latitude and tropical regions. The reason is that these ERs had been determined using an older set of tropopause heights, which had been calculated in a slightly different manner. In the update they are now consistently calculated based on the latest version of tropopause heights.

In Fig. A2 there are now white areas mainly above South-East Asia (no data due to clouds) for all gases, while this was the case for HCN only in the original manuscript. The reason is that the version of the program for calculation of the original global averages of C$_2$H$_6$, C$_2$H$_2$, PAN and HCOOH contained a small bug. This has now been corrected.

As requested by P. Shvedko (18 Jun 2024), the footnote "Note that the HCOOH distributions ..." on page 10 of the manuscript has been implemented within the main text.

Following the ACP guidelines, we now write "Fig." instead of "Figure" throughout the text, except at the beginning of sentences.

Following the ACP guidelines, we changed the unit "deg" into "°" in each plot of global distributions.